MIT-CTP-5594

August 21, 2023

# Open string stub as an auxiliary string field

Harold Erbin[1,2,3] and Atakan Hilmi Fırat[1,2]

[1] *Center for Theoretical Physics*
*Massachusetts Institute of Technology*
*Cambridge MA 02139, USA*

[2] *NSF AI Institute for Artificial Intelligence and Fundamental Interactions*

[3] *Université Paris Saclay, CEA, LIST*
*Gif-sur-Yvette, F-91191, France*

erbin@mit.edu, firat@mit.edu

**Abstract**

Witten's open string field theory with a generalized version of stubs is reformulated as a cubic string field theory using an auxiliary string field. The gauge symmetries and equations of motion as well as the associative algebra of the resulting theory are investigated. Integrating out either the original or auxiliary field is shown to recover the conventional cubic theory. Our analysis demonstrates that deformations due to the stubs can be described as a homotopy transfer purely in the context of strong deformation retract. We also discuss to what extent the vertex regions resulting from stubs provide a model for the elementary interactions of closed string field theory.

# 1 Introduction

String field theory (SFT) is a second-quantized formulation of string theory, for reviews see [1–5]. Despite the enormous success of open SFT to capture the non-perturbative physics of open string with its analytic solutions [6–13], the same hasn't been achieved for closed SFT so far due to its non-polynomial nature.[1] Solving closed SFT seems to require a set of novel perspectives, especially on the nature of string vertices describing the elementary interactions of closed strings—beyond the fact that they have to solve the geometric Batalin-Vilkovisky (BV) equation.

As a toy model of string vertices of closed SFT, the string vertices of the Witten's open SFT with stubs gained attraction recently due to its theoretical tractability [37], also see the past works [38–43]. For general SFT, given the local coordinates around the punctures $w_i(z)$ on a Riemann surface with

---

[1]We also point out the recent progress on D-instantons [14–29], conformal perturbation theory [30], the open-closed SFT [31–33], and understanding correlation functions from the perspective of homotopy algebras [34–36].

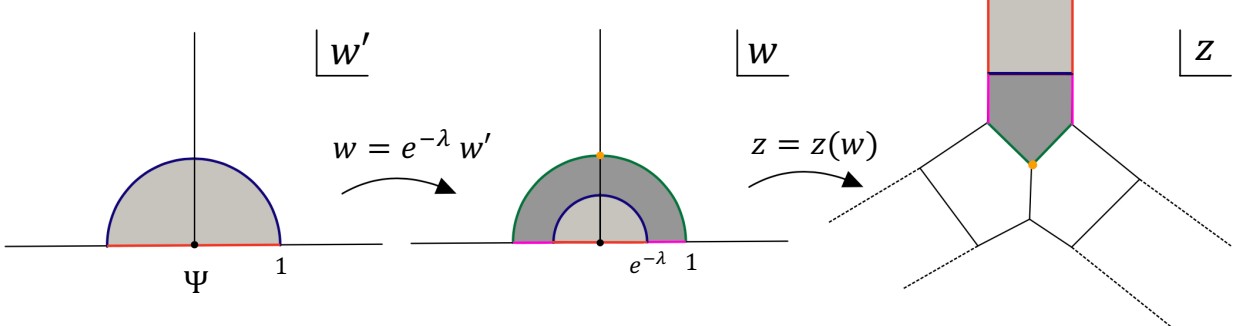

**Figure 1:** The Witten's vertex with stubs. We have used the same colors for the points/curves/regions that map each other. The local coordinates around a puncture are given by the half unit-disk in the $w'$-plane after adding stubs (light gray) and the string field $\Psi$ is placed at $w' = 0$ in these coordinates. The local coordinates for the remaining punctures work analogously.

an uniformizing coordinate $z$, adding a stub of length $\lambda$ (measured in the flat metric) amounts to using a new set of local coordinates (see figure 1)

$$w_i'(z) = e^\lambda w_i(z), \tag{1.1}$$

for $0 < |w_i'(z)| \le 1$, $i = 1, \cdots, n$. Including stubs enlarges the vertex region, i.e. it takes surfaces from the Feynman region and makes them part of the vertex region. In the case of Witten's open SFT, stubs induce vertex regions in the moduli spaces of disks with boundary punctures and the open SFT is no longer manifestly cubic—in fact it becomes non-polynomial. We call the resulting formulation *the non-polynomial stubbed open SFT*.

In this note we reformulate the stubbed open SFT as a manifestly cubic theory using a single auxiliary string field $\Sigma$ [44]. We demonstrate that its action can be given by

$$S[\Psi, \Sigma] = -\frac{1}{g_o^2} \left[ \frac{1}{2} \langle \Psi, Q_B \Psi \rangle + \frac{1}{2} \left\langle \Sigma, \frac{Q_B}{1 - e^{-2\lambda L_0}} \Sigma \right\rangle \right. \tag{1.2}$$

$$\left. - \frac{1}{3} \left\langle \Sigma - e^{-\lambda L_0} \Psi, \left( \Sigma - e^{-\lambda L_0} \Psi \right) * \left( \Sigma - e^{-\lambda L_0} \Psi \right) \right\rangle \right],$$

and it is endowed with the gauge transformations

$$\Psi \to \Psi + Q_B \Lambda_1 + e^{-\lambda L_0} \left[ \Sigma - e^{-\lambda L_0} \Psi, \Lambda_2 - e^{-\lambda L_0} \Lambda_1 \right], \tag{1.3a}$$

$$\Sigma \to \Sigma + Q_B \Lambda_2 - (1 - e^{-2\lambda L_0}) \left[ \Sigma - e^{-\lambda L_0} \Psi, \Lambda_2 - e^{-\lambda L_0} \Lambda_1 \right]. \tag{1.3b}$$

For definitions and conventions, see section 2.1. In fact, we show that this cubic construction not only works for the ordinary stubs (1.1), but also for their appropriate generalizations as we briefly mention in section 5. We remark that our use of additional string field is different from those in [45] and [46, 47], for which the additional field has been used to write a kinetic term involving Ramond sector fields in type II super SFT and to relax the level-matching constraint in the free bosonic closed SFT respectively. The auxiliary string field $\Sigma$ carries the same ghost number and Grassmannality with $\Psi$, while its kinetic term is rather unconventional.

Reader may wonder the point of introducing another string field to present the stubbed open SFT manifestly cubic way, given that we already have a convenient formulation in terms of Witten's

cubic theory. As we mentioned earlier, the vertex regions of the non-polynomial stubbed open SFT can be viewed as a toy model for the vertex regions of closed SFT and the motivation behind this study was to get an insight on whether introducing auxiliary string fields can cast the closed SFT to a manifestly cubic theory. We comment on what we have learned from our analysis in conclusion 6.

We note that the expectation of having a cubic closed SFT may not be as impossible as it initially sounds. Even though a no-go theorem restricts having a covariant cubic closed SFT [48], its underlying $L_\infty$ structure and *the strictification theorem* (or *rectification theorem*) of homotopy algebras demands that it should be still possible to recast it into a manifestly cubic formulation [49–51]. This indicates one of the assumptions of [48] has to be violated. The failed assumption is having a single string field in the theory: the strictification necessitates extending the Hilbert space, i.e. introducing auxiliary string fields. Integrating out these extra fields would then naturally result in the non-polynomial structure of the closed SFT in its current formulation.

However, this supposed cubic formulation may still not be amenable to practical calculations, such as finding solutions, as the complexity of the theory gets shifted from interactions to auxiliary fields. Nonetheless, the authors think, based on the recent developments in hyperbolic SFT [52–61], this may not be necessarily the case and this approach deserves further investigation. One of us has recently showed that the hyperbolic string vertices are intimately connected to Liouville theory and it is possible to construct a "bootstrap" program for string vertices [60]. The geometric data for the vertices is entirely encoded at the *cubic* level, given in terms of the semi-classical limit of the DOZZ formula, and the higher order interactions are characterized in terms of this cubic data and the classical conformal blocks [62–65]. However, this property of hyperbolic SFT is rather hidden and it would be desirable to make it manifest through a collection of auxiliary string fields. Another way to put is that it is conceivable that hyperbolic vertices may lead to *geometrization* of the strictification theorem for the homotopy algebras emanating from the moduli spaces of Riemann surfaces.

Beyond possible ramifications for closed SFT, our analysis further demonstrates that the deformations due to the stubs can be described as a homotopy transfer purely in the context of strong deformation retract thanks to the auxiliary string field. This is in contrast to [37], where the authors had to consider a situation for which this was not the case. Having a strong deformation retract clarifies many points in the algebraic manipulations and establishes a firmer ground for the procedure of including stubs in the homotopy algebraic framework. Nonetheless, the methods of [37] are perfectly sound and we argue our results are consistent with each other.

The rest of the paper is organized as follows. In section 2 we present our conventions for the Witten's open SFT and include stubs to it, which renders the theory non-polynomial. We introduce an auxiliary string field to make the theory cubic again and discuss the tachyon potential truncated to the lowest level in this theory as an example. In section 3, we investigate the gauge symmetries and equations of motion of the resulting theory as well as the consequences of integrating out fields. We show that there is an associative algebra underlying our theory in section 4 and then complete the discussion on integrating out fields using homological perturbation theory. We introduce the notion of generalized stubs in section 5 and discuss how our construction accommodates them. We conclude the paper in section 6. In appendix A we demonstrate that our algebraic procedure for including stubs is not specific to the Witten's theory, but can be applied to any theory based on an $A_\infty$ algebra. This allows us to argue for the equivalence of our procedure to [37] algebraically.

## 2 Open SFT with stubs

In this section we present our conventions for the Witten's open SFT and describe the procedure of adding stubs. This results in a non-polynomial open SFT. We recast this stubbed theory a cubic form by introducing an auxiliary string field. We investigate the tachyon potential truncated to the lowest level in this cubic theory and show that the depth of the potential (i.e. the tension of the associated D-brane) is equal to the one obtained from the Witten's theory truncated to the lowest level, suggesting our procedure of adding auxiliary string field is well-defined.

### 2.1 Witten's cubic open SFT

The Witten's cubic open (bosonic) SFT is given by ($\alpha' = 1$) [13, 66]

$$\widetilde{S}[\Psi] = -\frac{1}{g_o^2} \left[ \frac{1}{2} \langle \Psi, Q_B \Psi \rangle + \frac{1}{3} \langle \Psi, \Psi * \Psi \rangle \right], \tag{2.1}$$

where $g_o$ is the open string coupling constant. Here $\Psi \in \mathcal{H}$ is a ghost number 1 element of the Hilbert space of $c = 26$ matter $+$ $bc$ ghost boundary conformal field theory (BCFT) $\mathcal{H}$, $Q_B : \mathcal{H} \to \mathcal{H}$ is the BRST operator of $\mathcal{H}$, and the binary operation $* : \mathcal{H} \otimes \mathcal{H} \to \mathcal{H}$ is the star product defined by the Witten's vertex. The BRST operator $Q_B$ increases the ghost number by 1 while $*$ keeps the ghost number same. The BRST operator is nilpotent, $Q_B^2 = 0$. The bilinear form $\langle \cdot, \cdot \rangle$ is the BPZ inner product given in terms of the following BCFT correlator on the upper-half plane:

$$\langle A, B \rangle \equiv \langle (I \circ A(0)) B(0) \rangle. \tag{2.2}$$

Here $I(z) = -1/z$ is the inversion map. It satisfies

$$\langle A, B \rangle = (-1)^{|A||B|} \langle B, A \rangle, \qquad \langle A, B * C \rangle = \langle A * B, C \rangle, \tag{2.3}$$

where $|A|$ denotes the Grassmannality of $A$. The BPZ inner product is non-vanishing only if the ghost numbers of $A$ and $B$ add up to three by the ghost number anomaly. It is non-degenerate.

The BRST operator $Q_B$, together with the star product, forms a differential graded associative algebra. This amounts to the following identities

$$A * (B * C) = (A * B) * C, \qquad Q_B(A * B) = (Q_B A) * B + (-1)^{|A|} A * (Q_B B), \tag{2.4}$$

for any element $A, B \in \mathcal{H}$. The BRST operator $Q_B$ further satisfies

$$\langle A, Q_B B \rangle = (-1)^{|A|+1} \langle Q_B A, B \rangle. \tag{2.5}$$

We remark that this identity is true for any BPZ and Grassmann odd operator by definition, in particular it still holds when $Q_B \to c_0 L_0$. We also remark that the Grassmann even operator $e^{-\lambda L_0}$ for $\lambda \geq 0$ is BPZ even, i.e. it satisfies

$$\langle A, e^{-\lambda L_0} B \rangle = \langle e^{-\lambda L_0} A, B \rangle. \tag{2.6}$$

The equation of motion and the gauge symmetry of the action (2.1) are given by

$$Q_B \Psi + \Psi * \Psi = 0, \qquad \Psi \to \Psi + Q_B \Lambda + \Psi * \Lambda - \Lambda * \Psi \equiv \Psi + Q_B \Psi + [\Psi, \Lambda]. \tag{2.7}$$

Here $\Lambda$ is a gauge parameter and it carries ghost number 0. The gauge symmetry can be used to set the theory to the Siegel gauge, $b_0 \Psi = 0$, where $b_0$ is the zero mode of the $b$-ghost. This gives rise to the gauge-fixed theory

$$\widetilde{S}[\Psi] = -\frac{1}{g_o^2} \left[ \frac{1}{2} \langle \Psi, c_0 L_0 \Psi \rangle + \frac{1}{3} \langle \Psi, \Psi * \Psi \rangle \right] , \tag{2.8}$$

where $c_0, L_0$ are the zero modes of the $c$-ghost and the stress energy tensor $T(z)$ of the BCFT respectively. The propagator in this gauge is given by

$$\Delta \equiv \frac{b_0}{L_0} = b_0 \int\limits_0^\infty ds \, e^{-sL_0} , \tag{2.9}$$

where the quantity $s$ can be interpreted as the proper length of the propagating string. This is because the operator $e^{-sL_0}$ corresponds to the operation of gluing flat strip of length $s$ to the Witten's vertex, see figure 1. Feynman diagrams are constructed by gluing Witten's vertices together with the flat strips.

## 2.2 Inclusion of stubs and the auxiliary string field

We now add stubs of length $\lambda$ to the gauge-fixed theory (2.8). This modifies the action by

$$-g_o^2 \, S[\Psi] = \frac{1}{2} \langle \Psi, c_0 L_0 \Psi \rangle + \frac{1}{3} \langle e^{-\lambda L_0} \Psi, e^{-\lambda L_0} \Psi * e^{-\lambda L_0} \Psi \rangle + \cdots . \tag{2.10}$$

Here dots stand for the terms that get induced by the higher elementary vertices appearing in the stubbed theory. For example, the quartic term is given by

$$-g_o^2 \, S[\Psi] \supset \frac{1}{4} (-2) \left\langle e^{-\lambda L_0} \Psi * e^{-\lambda L_0} \Psi , \left[ b_0 \int_0^{2\lambda} ds \, e^{-sL_0} \right] \left( e^{-\lambda L_0} \Psi * e^{-\lambda L_0} \Psi \right) \right\rangle . \tag{2.11}$$

The logic behind this term is as follows. The vertex regions consist of the Feynman diagrams of the Witten's theory whose propagating strings have proper length smaller than $2\lambda$ after including stubs of length $\lambda$. Considering the color-ordered diagrams, there is $1/4$ in front of the quartic term by the cyclic symmetry and $-2$ results from including the missing Feynman diagrams as vertices. We remark that the string propagator whose proper length is bounded by $2\lambda$ evaluates to

$$b_0 \int\limits_0^{2\lambda} ds \, e^{-sL_0} = b_0 \, \frac{1 - e^{-2\lambda L_0}}{L_0} , \tag{2.12}$$

and this appears in the term (2.11) by the form of the vertex regions explained above. Notice its inverse in the Siegel gauge is given by

$$\left[ b_0 \int\limits_0^{2\lambda} ds \, e^{-sL_0} \right]^{-1} = \frac{c_0 \, L_0}{1 - e^{-2\lambda L_0}} , \tag{2.13}$$

Now we introduce an auxiliary string field $\Sigma \in \mathcal{H}$ with ghost number 1 and include the following additional term to the action

$$-g_o^2 \, \delta S[\Psi, \Sigma] = \frac{1}{2} \left\langle \Sigma - b_0 \frac{1 - e^{-2\lambda L_0}}{L_0} \left( e^{-\lambda L_0} \Psi * e^{-\lambda L_0} \Psi \right) , \right. \tag{2.14}$$

$$\left. \frac{c_0 \, L_0}{1 - e^{-2\lambda L_0}} \left[ \Sigma - b_0 \frac{1 - e^{-2\lambda L_0}}{L_0} \left( e^{-\lambda L_0} \Psi * e^{-\lambda L_0} \Psi \right) \right] \right\rangle .$$

This term is non-vanishing by the identities (2.5) and (2.6). The equation of motion of the field $\Sigma$ is given by

$$\Sigma - b_0 \frac{1 - e^{-2\lambda L_0}}{L_0} \left( e^{-\lambda L_0} \Psi * e^{-\lambda L_0} \Psi \right) = 0 . \qquad (2.15)$$

Notice $\Sigma$ is determined in terms of $\Psi$ and we recover the stubbed theory (2.11) upon inserting the equation of motion of $\Sigma$ to the extra term (2.14) (i.e. upon integrating $\Sigma$ out). Hence it is justified to call $\Sigma$ an auxiliary field. We additionally see $\Sigma$ satisfies $b_0 \Sigma = 0$. This is going to be interpreted as the gauge-fixing condition for the auxiliary field $\Sigma$ in the Siegel gauge eventually. We discuss the gauge-invariant theory in section 3.

Combining two terms (2.11) and (2.14) we are lead to the combined action

$$-g_o^2 \, S[\Psi, \Sigma] = -g_o^2 \left( S[\Psi] + \delta S[\Psi, \Sigma] \right) = \frac{1}{2} \langle \Psi, c_0 L_0 \Psi \rangle + \frac{1}{3} \langle e^{-\lambda L_0} \Psi \, , e^{-\lambda L_0} \Psi * e^{-\lambda L_0} \Psi \rangle \qquad (2.16)$$

$$+ \frac{1}{2} \left\langle \Sigma \, , \frac{c_0 \, L_0}{1 - e^{-2\lambda L_0}} \Sigma \right\rangle - \left\langle \Sigma \, , e^{-\lambda L_0} \Psi * e^{-\lambda L_0} \Psi \right\rangle + \cdots ,$$

where we have used the identities (2.5) and (2.6). We see that the (truncated) theory is in a cubic form now and the quadratic term is compensated by having an auxiliary field $\Sigma$. Here we essentially performed a Hubbard-Stratonovich transformation for the quartic term [67].

Notice the propagator of $\Sigma$ does not have a pole as $L_0 \to 0$ since

$$b_0 \frac{1 - e^{-2\lambda L_0}}{L_0} = 2\lambda \, b_0 \left[ 1 - \lambda \, L_0 + \cdots \right] , \qquad (2.17)$$

as it should be for an auxiliary field. This expansion further implies that the kinetic term of $\Sigma$ is well-defined for any $\Sigma \in \mathcal{H}$. We remark that an alternative way to see $\Sigma$ is auxiliary is to consider the equation of motion for $\Sigma$ for the free theory

$$\frac{c_0 \, L_0}{1 - e^{-2\lambda L_0}} \Sigma = \left[ \frac{1}{2\lambda} + \frac{L_0}{2} + \cdots \right] c_0 \, \Sigma = 0, \qquad (2.18)$$

and notice the only solution when $L_0 = 0$ (i.e. when on-shell) reads $\Sigma = 0$ given that $b_0 \, \Sigma = 0$. We point out that these statements hold true when $b_0 \, \Sigma = 0$ and they may be violated when it is no longer imposed (i.e. *gauge-unfixed*). This would require introducing a certain constraint on $\Sigma$ and the gauge transformation. We further elaborate on this point in section 3.

Before we investigate the stubbed theory further, let us generalize the procedure above to higher orders. This generalization can be achieved upon demanding that the auxiliary string field $\Sigma$, and its interactions, are there to compensate for the surfaces with internal flat strips of length smaller than $2\lambda$ that would have got missed in the non-polynomial stubbed theory without including higher-order vertices. Notice this was precisely what has happened for the quartic interaction, refer to the action (2.16) and its Feynman diagrams also figure 2.

This reasoning implies that it is sufficient to include cubic couplings of the form $\Sigma^2 \Psi$ and $\Sigma^3$ to cover the moduli space of disks with boundary punctures given that Witten's theory provide a single cover for them already [68]—we don't need any more auxiliary string fields. As a result, the

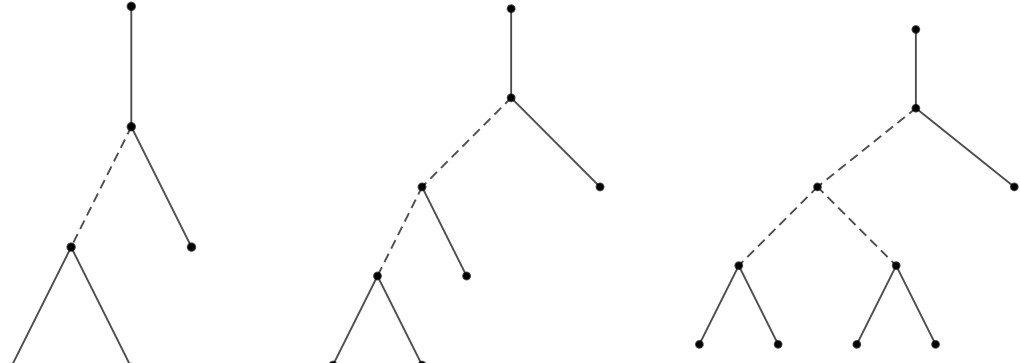

**Figure 2:** Examples of full binary trees corresponding to the 4-,5-and 6-point string diagrams to demonstrate the types of interactions needed between the string fields $\Psi$ and $\Sigma$. We remind that there is no propagator associated with the external edges.

stubbed open SFT can be written in the following manifestly cubic form

$$-g_o^2 S[\Psi,\Sigma] = \frac{1}{2}\langle\Psi,c_0 L_0\Psi\rangle + \frac{1}{3}\langle e^{-\lambda L_0}\Psi\,,e^{-\lambda L_0}\Psi * e^{-\lambda L_0}\Psi\rangle \tag{2.19}$$

$$+ \frac{1}{2}\left\langle\Sigma\,,\frac{c_0\,L_0}{1-e^{-2\lambda L_0}}\Sigma\right\rangle - \left\langle\Sigma\,,e^{-\lambda L_0}\Psi * e^{-\lambda L_0}\Psi\right\rangle + \langle e^{-\lambda L_0}\Psi,\Sigma * \Sigma\rangle - \frac{1}{3}\langle\Sigma,\Sigma * \Sigma\rangle$$

$$= \frac{1}{2}\langle\Psi,c_0 L_0\Psi\rangle + \frac{1}{2}\left\langle\Sigma\,,\frac{c_0\,L_0}{1-e^{-2\lambda L_0}}\Sigma\right\rangle - \frac{1}{3}\left\langle\Sigma - e^{-\lambda L_0}\Psi,\left(\Sigma - e^{-\lambda L_0}\Psi\right) * \left(\Sigma - e^{-\lambda L_0}\Psi\right)\right\rangle.$$

We point out the signs are induced by the sign of the $\Psi^3$ term and the coefficients are due to the cyclicity of diagrams. Like for the quartic term above, these are the consequences of considering color-ordered diagrams. This action is real assuming $\Sigma$ carries the same reality properties as $\Psi$.

Let us demonstrate the argument above graphically by considering the full binary trees as a model of (color-ordered) Feynman diagrams of open strings, a set of relevant examples are shown in figure 2. Recall a tree is an undirected graph for which vertices are connected exactly once and a rooted tree is a specific tree for which one vertex is designated as root. An ordered tree is a rooted tree for which there is an ordering among "daughters" of each vertex. Daughter in this context means a vertex connected to the "parent" vertex that is not on the path to the root. Finally, full binary tree is an ordered tree for which each vertex has only two daughters or no daughters at all. We call the vertices without daughters "leafs". We represent the external strings by the root and leafs of the tree, the propagators of $\Psi$ by solid internal edges, and the (stubbed) $\Psi^3$ interactions by the vertices for which three solid edges connect. The propagators of $\Sigma$ is represented by dashed edges. We associate no propagators with the external edges.

Full binary trees with solid edges don't lead to a full cover of the moduli spaced due to the stubs—inclusion of dashed vertices are necessary. In (2.16), we have included the $\Sigma\Psi^2$ interaction (i.e. dashed-solid-solid vertex) and this was sufficient to eliminate the quartic vertex. However, the binary trees with more leafs allow for the possibility of having $\Sigma^2\Psi$ and $\Sigma^3$ interactions (i.e. dashed-dashed-solid and dashed-dashed-dashed vertices), see figure 2. It is necessary to include them to cover the moduli spaces entirely. Beyond them, we don't need to include further interactions and/or auxiliary fields by the correspondence between the full binary trees and color-ordered Feynman diagrams. This justifies the sole appearance of the $\Sigma^2\Psi$ and $\Sigma^3$ interactions in the action (2.19).

In summary, we recast the stubbed open SFT into a cubic form with an auxiliary string field. It is interesting to notice in the $\lambda \to 0$ limit, i.e. when there are no stubs, the kinetic term of the string field $\Sigma$ blows up, or equivalently the propagator for $\Sigma$ vanishes. This is expected: the propagation of the field $\Sigma$ should vanish as there are no vertex regions to compensate in this case. In this case the equations of motion set $\Sigma = 0$, which doesn't change as $\Psi$ changes, so $\Sigma$ can be ignored for all intents and purposes.

It is also interesting to investigate the infinite stub limit, $\lambda \to \infty$. For states with $L_0 > 0$, the string fields $\Psi$ and $\Sigma$ decouple: $\Psi$ describes a free SFT and $\Sigma$ describes the Witten's theory after the field redefinition $\Sigma \to -\Sigma$. This makes sense: for infinite stubs the physics of the massive states transfers *entirely* to the field $\Sigma$ while the original field $\Psi$ is left with no dynamics associated with these modes. This is because $\Sigma$ captures the small proper length open string propagation by construction, where the higher $L_0$ modes are the most relevant. When $L_0 \leq 0$, on the other hand, the kinetic term for $\Sigma$ vanishes as $\lambda \to \infty$ and the interactions of $\Sigma$ are suppressed relative to the $\Psi^3$ interaction. The remaining theory is effectively the Witten's theory described by $\Psi$. We remark that having stubs amplifies the interactions of the tachyons compared to the rest of the states.

## 2.3 The tachyon potential truncated to the lowest level

In order to provide an evidence for the action (2.19) contains the same physics as the Witten's theory, let us evaluate the tachyon potential truncated to the lowest level fields, i.e. $L_0 = -1$, for the critical bosonic theory and show that the depths of the potential computed using the action (2.1) and the action with the auxiliary field (2.19) are the same. We reserve the general discussion on matching on-shell actions in both theories to section 4 after we develop some algebraic machinery.

So, begin by considering a D$p$-brane wrapping $p+1$ compact dimensions. Its mass $M$ is related to the open string coupling constant $g_o$ through $M = 1/(2\pi^2 g_o^2)$ [6]. The tachyon potential $\widetilde{V}$ of the Witten's theory truncated to the lowest level is famously given by [7]

$$\widetilde{V} = -\widetilde{S} \implies \frac{\widetilde{V}}{2\pi^2 M^2} = -\frac{1}{2}t^2 + \frac{1}{3}\frac{t^3}{r^3}, \qquad r = \frac{4}{3\sqrt{3}}, \tag{2.20}$$

using $\Psi = t\, c_1 |0\rangle + \ldots$ in (2.8). Here $|0\rangle$ is the $PSL(2,\mathbb{R})$ invariant vacuum, $t$ is the zero momentum tachyon field, and $r$ is the notorious constant of SFT. The place and the depth of the vacuum are

$$t_* = r^3 = \left(\frac{4}{3\sqrt{3}}\right)^3 \quad \text{and} \quad \frac{\widetilde{V}(t_*)}{M^2} = -\frac{\pi^2}{3}r^6 = -\frac{4096\pi^2}{59046} \approx -0.684, \tag{2.21}$$

which produces $\approx \%68$ of the correct answer $V(t_*) = -M^2$.

Now, we consider the tachyon potential of the theory with the auxiliary field $\Sigma$ (2.19). The truncated tachyon potential in this case is given by

$$V = -S \implies \frac{V}{2\pi^2 M^2} = -\frac{1}{2}t^2 - \frac{1}{2}\frac{\sigma^2}{1 - e^{2\lambda}} - \frac{1}{3}\frac{(\sigma - e^\lambda t)^3}{r^3}, \tag{2.22}$$

where we truncated $\Sigma = \sigma c_1 |0\rangle + \cdots$ to $L_0 = -1$ similar to $\Psi$. Here the component field $\sigma$ is a zero momentum auxiliary bosonic scalar field in the target space. The vacuum is now at

$$t_* = e^\lambda r^3 \quad \text{and} \quad \sigma_* = 2e^\lambda \sinh(\lambda)\, r^3, \tag{2.23}$$

while the depth of the potential is still given by

$$\frac{V(t_*, \sigma_*)}{M^2} = -\frac{\pi^2}{3} r^6 \,.$$

(2.24)

Even though the depth of the potential hasn't changed, the position of the vacuum moves exponentially far away in the field space. This is a reflection of the fact that the stubs make the non-perturbative physics difficult to access, see the discussion in [3]. We point out that the level truncation schemes we used in both theories are the same as we truncate both $\Psi$ *and* $\Sigma$ to $L_0 = -1$ in the Siegel gauge. This analysis can be extended to higher levels straightforwardly.

# 3 The cubic stubbed open SFT

Our primary goal in this section is to obtain the gauge symmetries and equations of motion of the action (2.19) after gauge-unfixing. We then investigate ways to integrate out the fields $\Sigma$ and $\Psi$ from the theory. As we shall demonstrate, we get the Witten's cubic open SFT after integrating out either string fields—more precisely, after integrating out a *certain* combination of them. This was expected for the string field $\Sigma$. Somewhat surprisingly, we also find that the theory reduces to the Witten's theory even when one integrates $\Psi$ out, leaving the auxiliary field $\Sigma$.

## 3.1 The gauge symmetries and equations of motion

We begin by gauge-unfixing the action (2.19). Our proposal for the gauge-invariant action is

$$-g_o^2 S[\Psi, \Sigma] = \frac{1}{2} \langle \Psi, Q_B \Psi \rangle + \frac{1}{2} \left\langle \Sigma \,, \frac{Q_B}{1 - e^{-2\lambda L_0}} \Sigma \right\rangle$$

$$- \frac{1}{3} \left\langle \Sigma - e^{-\lambda L_0} \Psi, \left( \Sigma - e^{-\lambda L_0} \Psi \right) * \left( \Sigma - e^{-\lambda L_0} \Psi \right) \right\rangle \,,$$

(3.1)

whose gauge transformations are given by

$$\Psi \to \Psi + Q_B \Lambda_1 + e^{-\lambda L_0} \left[ \Sigma - e^{-\lambda L_0} \Psi \,, \Lambda_2 - e^{-\lambda L_0} \Lambda_1 \right] \,,$$

(3.2a)

$$\Sigma \to \Sigma + Q_B \Lambda_2 - (1 - e^{-2\lambda L_0}) \left[ \Sigma - e^{-\lambda L_0} \Psi \,, \Lambda_2 - e^{-\lambda L_0} \Lambda_1 \right] \,,$$

(3.2b)

where $\Lambda_1, \Lambda_2 \in \mathcal{H}$ are distinct ghost number 0 states. It is easy to directly check gauge transformations leave the action above invariant. We note that the kinetic term does not seem to be well-defined if $\Sigma \in \ker L_0$ in (3.1), despite it being well-defined in the "gauge-fixed" form in (2.19). The simplest solution to avoid this problem after gauge-unfixing is to take $\Sigma \in \mathcal{H} \cap (\ker L_0)^c$ from the get go. This also requires $\Lambda_2 \in \mathcal{H} \cap (\ker L_0)^c$. As we shall see, no point of the argument needs $\Sigma$ or $\Lambda_2$ to contain $L_0 = 0$ modes and that's what we assume this moment forward.

The logic behind this gauge-unfixing is as follows. It is expected that $c_0 L_0 \to Q_B$ in the kinetic terms when we gauge-unfix from the Siegel gauge. Therefore it is natural to make the following ansatz for the gauge symmetry for the $c_0 L_0 \to Q_B$ replaced action:

$$\Psi \to \Psi + Q_B \Lambda_1 + \cdots \,, \qquad \Sigma \to \Sigma + Q_B \Lambda_2 + \cdots \,.$$

(3.3)

Here $\Lambda_1$ and $\Lambda_2$ are distinct string fields and dots stand for the possible field-dependent part of the gauge transformation. The ansatz (3.3) would have worked only if the interaction terms were absent—under (3.3) the interaction terms transform as

$$\delta_{\Lambda_1,\Lambda_2} S[\Psi,\Sigma] = \frac{1}{g_o^2} \left\langle Q_B(\Lambda_2 - e^{-\lambda L_0}\Lambda_1), \left(\Sigma - e^{-\lambda L_0}\Psi\right) * \left(\Sigma - e^{-\lambda L_0}\Psi\right)\right\rangle, \tag{3.4}$$

for which we have used properties of the BPZ inner product. This term has to be compensated by the field-dependent terms in the gauge transformations (3.2). The form of $\delta_{\Lambda_1,\Lambda_2} S[\Psi,\Sigma]$ entirely fixes the field-dependent terms as in (3.2), mirroring the situation for the Witten's theory (2.7).

As a consistency check, we can show that the Siegel gauge $(b_0 \Psi = b_0 \Sigma = 0)$ is reachable using (3.2). For this, we first observe the following gauge transformation:

$$e^{-\lambda L_0}\Psi - \Sigma \to e^{-\lambda L_0}\Psi - \Sigma + Q_B\left(e^{-\lambda L_0}\Lambda_1 - \Lambda_2\right) + \left[e^{-\lambda L_0}\Psi - \Sigma, e^{-\lambda L_0}\Lambda_1 - \Lambda_2\right]. \tag{3.5}$$

This implies we can set $b_0\left(\Sigma - e^{-\lambda L_0}\Psi\right) = 0$ by adjusting the combination $\Lambda_2 - e^{-\lambda L_0}\Lambda_1$ just as in the Witten's theory. Now the problem reduces to showing whether fields can be fixed to Siegel gauge individually. We define

$$\Xi \equiv \left[\Sigma - e^{-\lambda L_0}\Psi, \Lambda_2 - e^{-\lambda L_0}\Lambda_1\right], \tag{3.6}$$

and we have

$$b_0\Psi' = b_0\left(\Psi + Q_B\Lambda_1 + e^{-\lambda L_0}\Xi\right), \tag{3.7}$$

after the gauge transformation $\Psi \to \Psi'$. We would like to show there exist gauge parameters such that this equation can be set to 0. We consider this gauge transformation after fixing the combination $\Sigma - e^{-\lambda L_0}\Psi$ to the Siegel gauge $b_0\left(\Sigma - e^{-\lambda L_0}\Psi\right) = 0$.

Using $\{Q_B, \Delta\} = 1$ with $\Delta = b_0/L_0$ we notice[2]

$$b_0\{Q_B, \Delta\}\left(\Psi + e^{-\lambda L_0}\Xi\right) = b_0 Q_B \Delta\left(\Psi + e^{-\lambda L_0}\Xi\right), \tag{3.8}$$

upon using $b_0\Delta = 0$. We then have

$$b_0\Psi' = b_0 Q_B\left[\Lambda_1 + \Delta\left(\Psi + e^{-\lambda L_0}\Xi\right)\right]. \tag{3.9}$$

In order to set this to zero we must have $\Lambda_1 + \Delta\left(\Psi + e^{-\lambda L_0}\Xi\right) \in \ker b_0 Q_B$. We can simply take the gauge transformation to be

$$\Lambda_1 = -\Delta\left(\Psi + e^{-\lambda L_0}\Xi\right), \tag{3.10}$$

up to a term in $\ker b_0 Q_B$. A similar reasoning shows

$$\Lambda_2 = -\Delta\left(\Sigma - (1 - e^{-2\lambda L_0})\Xi\right), \tag{3.11}$$

up to a term in $\ker b_0 Q_B$. For the argument below it is sufficient to take the possible term in $\ker b_0 Q_B$ to be zero.

---

[2]The argument breaks down for $L_0 = 0$ in a way how Feynman gauge breaks down on-shell in gauge theories. Nonetheless, the subtleties associated with this situation can be dealt exactly analogous to the gauge theories, see [4].

Notice previous two equations are coupled to each other given that $\Xi = \Xi(\Lambda_1, \Lambda_2)$. However, it is possible to solve these equations simultaneously by taking

$$\Lambda_1 = -\Delta\Psi \quad \text{and} \quad \Lambda_2 = -\Delta\Sigma\,, \tag{3.12}$$

given that we have

$$\Xi = -[\Sigma - e^{-\lambda L_0}\Psi\,, \Delta\,(\Sigma - e^{-\lambda L_0}\Psi)] = 0\,, \tag{3.13}$$

using the definition (3.6) and $b_0\,(\Sigma - e^{-\lambda L_0}\Psi) = 0$, which clearly doesn't get modified by this new gauge transformation, see (3.5). We conclude the Siegel gauge is reachable. Moreover, the gauge is fixed completely in the Siegel gauge, which can be argued similar to the Witten's theory [4].

This is a good point to comment more on the cohomology and physical states of the combined system $(\Psi, \Sigma)$. In the Siegel gauge, it was clear that the only physical states were the usual ones associated with $\Psi$ since the propagator of $\Sigma$ didn't have any pole, see (2.17). This fact stays true after gauge-unfixing given that we restrict $\Sigma$ to lie on the complement of $\ker L_0$ in order to define the kinetic term properly. Having $\Sigma$ belong to this complement implies that the operator $\Delta = b_0/L_0$ is well-defined acting on this part of the Hilbert space and satisfies $\{Q_B, \Delta\} = 1$, i.e. it is a *contracting homotopy operator* for $Q_B$ [4]. Since the cohomology can only live in the subspace where this operator is not well-defined, we see that the part associated with $\Sigma$ doesn't "double" the cohomology and give rise to additional physical states. So it is still justified to call $\Sigma$ an auxiliary field. The physical states in general can be described by some combination of $\Psi$ and $\Sigma$, nonetheless they are same as the ones in the original theory.

Finally, the equations of motion are given by

$$E_\Psi \equiv Q_B\Psi + e^{-\lambda L_0}\left[\left(\Sigma - e^{-\lambda L_0}\Psi\right) * \left(\Sigma - e^{-\lambda L_0}\Psi\right)\right] = 0\,, \tag{3.14a}$$

$$E_\Sigma \equiv \frac{Q_B}{1 - e^{-2\lambda L_0}}\Sigma - \left[\left(\Sigma - e^{-\lambda L_0}\Psi\right) * \left(\Sigma - e^{-\lambda L_0}\Psi\right)\right] = 0\,. \tag{3.14b}$$

upon varying (3.1).

## 3.2 Integrating out the auxiliary string field $\Sigma$

Our aim in this subsection is to describe different ways to integrate out the auxiliary string field $\Sigma$ from the theory. Begin by noticing the following combination of equations of motion implies

$$E_\Psi + e^{-\lambda L_0}E_\Sigma = Q_B\left[\Psi + \frac{e^{-\lambda L_0}}{1 - e^{-2\lambda L_0}}\Sigma\right] = 0 \implies \Psi + \frac{e^{-\lambda L_0}}{1 - e^{-2\lambda L_0}}\Sigma \in \ker Q_B\,. \tag{3.15}$$

In fact the resulting state is not just $Q_B$-closed but also can be taken zero after *partially* fixing a combination of the fields to the Siegel gauge via the gauge transformation (3.2)[3]

$$\Psi + \frac{e^{-\lambda L_0}}{1 - e^{-2\lambda L_0}}\Sigma \to \Psi + \frac{e^{-\lambda L_0}}{1 - e^{-2\lambda L_0}}\Sigma + Q_B\left[\Lambda_1 + \frac{e^{-\lambda L_0}}{1 - e^{-2\lambda L_0}}\Lambda_2\right]\,. \tag{3.16}$$

---

[3]Since we aim to integrate out a field that has a non-trivial transformation under the gauge symmetry a partial fixing of this form is convenient. Although this is not strictly necessary.

This procedure involves choosing a combination of $\Lambda_1, \Lambda_2$ appropriately. More explicitly, given $\Psi$ and $\Sigma$, the gauge parameters have to be chosen such that

$$\Lambda_1 + \frac{e^{-\lambda L_0}}{1 - e^{-2\lambda L_0}}\Lambda_2 = -\Delta\left[\Psi + \frac{e^{-\lambda L_0}}{1 - e^{-2\lambda L_0}}\Sigma\right]. \tag{3.17}$$

Doing this, the combination of the fields above can be taken to be annihilated by $b_0$ and we see

$$\Psi + \frac{e^{-\lambda L_0}}{1 - e^{-2\lambda L_0}}\Sigma \in \ker L_0 \implies (1 - e^{-2\lambda L_0})\Psi + e^{-\lambda L_0}\Sigma = 0$$

$$\implies \Sigma = -2\sinh(\lambda L_0)\Psi, \tag{3.18}$$

using the relation $\{Q_B, b_0\} = L_0$ and the fact that $1 - e^{-2\lambda L_0}$ vanishes on $\ker L_0$. Like in [37], we point out the operator $\sinh(\lambda L_0)$ may not make sense on the worldsheet level. Nevertheless, it is meaningful when the string field is expanded in the eigenstates of $L_0$ as long as they are finite. We assume $\sinh(\lambda L_0)$ (more generally $e^{\lambda L_0}$) to be a well-defined operation on the string fields.

In order to integrate out $\Sigma$ from the theory *non-perturbatively,*[4] we insert the equation (3.18) to the action (3.1) and evaluate. After using the identity (2.6) for the BPZ product, we get

$$-g_o^2\, S[\Psi, -2\sinh(\lambda L_0)\Psi] = \frac{1}{2}\langle e^{\lambda L_0}\Psi, Q_B\, e^{\lambda L_0}\Psi\rangle + \frac{1}{3}\langle e^{\lambda L_0}\Psi, e^{\lambda L_0}\Psi * e^{\lambda L_0}\Psi\rangle = -g_o^2\, \widetilde{S}[e^{\lambda L_0}\Psi]. \tag{3.19}$$

which is just the Witten's theory up to a field redefinition. The gauge transformation for $\Psi$ (3.2a) also reduces to

$$\Psi \to \Psi + Q_B\,\Lambda_1 + e^{-\lambda L_0}\left[\Sigma - e^{-\lambda L_0}\Psi, \, \Lambda_2 - e^{-\lambda L_0}\Lambda_1\right] \tag{3.20}$$

$$= \Psi + Q_B\,\Lambda_1 + e^{-\lambda L_0}\left[e^{\lambda L_0}\Psi, e^{\lambda L_0}\Lambda_1\right],$$

with the choice (3.17) and using (3.18). After redefining the gauge parameter $\Lambda_1 \to e^{-\lambda L_0}\Lambda_1$ we indeed get the gauge transformation of the Witten's theory (2.7). These were expected. In section 4 we describe the same procedure in the language of homotopy transfer.

It is also possible to integrate out the field $\Sigma$ *perturbatively* in order to obtain the non-polynomial stubbed open SFT of [37]. To that end, we first rescale $\Psi \to g_o\Psi$ and $\Sigma \to g_o\Sigma$ in order to keep track of the order of perturbation. Then we consider the expansion of $\Sigma$ in $g_o$

$$\Sigma = \sum_{n=1}^{\infty} g_o^n\, \Sigma^{(n)}, \tag{3.21}$$

plug it into the equation (3.14b), and solve the equation perturbatively in $g_o$. Observe that this procedure is different from the one described above: we insert different equation of motion to the action and evaluate it as a series in $g_o$, hence *perturbative.*

Notice we begin the series for $\Sigma$ at $n = 1$ because we would like to set $\Sigma = 0$ when $g_o = 0$ given that the theory should be free open SFT described by $\Psi$. Furthermore we take $b_0\,\Sigma = 0$ for our

---

[4]This is non-perturbative in the sense that the equation of motion we use to integrate out fields is independent of the string coupling constant $g_o$.

purposes.[5] As a result, we obtain the following recursion relation for $\Sigma^{(n)}$

$$\frac{Q_B}{1 - e^{-2\lambda L_0}} \Sigma^{(n)} - \sum_{k=0}^{n-1} \Sigma^{(k)} * \Sigma^{(n-k-1)} = 0 \,, \tag{3.22}$$

when $n \geq 1$ upon plugging the series (3.21) to the equation of motion (3.14b). Here we have defined $\Sigma^{(0)} = -e^{-\lambda L_0}\Psi$ for brevity. First few equations, together with their associated order in $g_o$, are given by

$$\mathcal{O}(g_o): \quad \frac{Q_B}{1 - e^{-2\lambda L_0}} \Sigma^{(1)} - e^{-\lambda L_0}\Psi * e^{-\lambda L_0}\Psi = 0 \,, \tag{3.23a}$$

$$\mathcal{O}(g_o^2): \quad \frac{Q_B}{1 - e^{-2\lambda L_0}} \Sigma^{(2)} + \Sigma^{(1)} * e^{-\lambda L_0}\Psi + e^{-\lambda L_0}\Psi * \Sigma^{(1)} = 0 \,. \tag{3.23b}$$

$$\vdots \qquad\qquad\qquad \vdots$$

It is possible to solve the recursion relation (3.22). At $\mathcal{O}(g_o)$, for instance, we find

$$\Sigma^{(1)} = (1 - e^{-2\lambda L_0})\Delta\left(e^{-\lambda L_0}\Psi * e^{-\lambda L_0}\Psi\right) = b_0 \frac{1 - e^{-2\lambda L_0}}{L_0}\left(e^{-\lambda L_0}\Psi * e^{-\lambda L_0}\Psi\right) \,, \tag{3.24}$$

after multiplying the equation (3.23a) with $b_0(1 - e^{-\lambda L_0})/L_0$ and using $\{Q_B, \Delta\} = 1$ and $b_0\Sigma = 0$. Notice $L_0 = 0$ modes don't cause any problem in this operation by the constraint $\Sigma \in (\ker L_0)^c$. Inserting $\Sigma^{(1)}$ into (2.19), we get

$$S[\Psi] = -\frac{1}{2}\langle\Psi, Q_B\Psi\rangle - \frac{g_o}{3}\langle e^{-\lambda L_0}\Psi, e^{-\lambda L_0}\Psi * e^{-\lambda L_0}\Psi\rangle \tag{3.25}$$

$$+ \frac{g_o^2}{2}\left\langle e^{-\lambda L_0}\Psi * e^{-\lambda L_0}\Psi \,, b_0\frac{1 - e^{-2\lambda L_0}}{L_0}\left(e^{-\lambda L_0}\Psi * e^{-\lambda L_0}\Psi\right)\right\rangle + \cdots \,.$$

Upon reabsorbing $g_o$ into the fields we indeed see this gives the combination of (2.10) and (2.11).

We postpone showing the generalization of this procedure to higher orders in $g_o$ after we introduce the algebraic structure associated with the action (3.1). Before that, we discuss the possible obstructions in solving the recursion (3.22). For example, at the leading order, the absence of obstruction requires

$$e^{-\lambda L_0}\Psi * e^{-\lambda L_0}\Psi \quad \text{is BRST exact,}$$

which is not true for generic $\Psi$. We can see that this term being BRST exact is a consequence of the equation of motion $E_\Psi$ (3.14a) after applying $Q_B$ to its $\mathcal{O}(g_o^0)$ equation. Even though this equation cannot be imposed completely, given that $\Psi$ would be a classical solution whereas we would like to keep it off-shell in the action, we are still allowed use it to show that the term above is BRST exact. The reason is that, as we can observe from our discussion on the gauge-fixing above, it is not possible to gauge fix $\Sigma$ independently of $\Psi$. Since we fix the gauge for $\Sigma$ while solving (3.22), we must also impose gauge fixing constraints for $\Psi$. In a sense this is like imposing the "Gauss' law" in this context.

Another way to see this is to keep the constraint aside from the action (3.25): since it is automatically satisfied when $\Psi$ is a solution to the equations of motion coming from (3.25), they do not

---

[5]This is not strictly necessary in order to solve the equations, however only this situation gives rise to the non-polynomial stubbed theory of [37].

constrain the dynamics. This is similar to what happens to the out-of-Siegel gauge constraints after gauge fixing and integrating out heavy fields [69]. Alternatively, we could explicitly decompose the BRST operator in terms of zero-modes [69] and use the constraint obtained from the $\Psi$ equation of motion to show that there is no obstruction. It is apparent that a similar argument holds at higher orders and we have no obstructions. This makes perfect sense—integrating out $\Sigma$ shouldn't get obstructed if it is really an auxiliary field.

### 3.3 Integrating out the original string field $\Psi$

We can repeat the procedure described above to integrate out the original string field $\Psi$ as well. We integrate out $\Psi$ non-perturbatively using the equation (3.18) with the same gauge choice. The action (3.1) now takes the form

$$-g_o^2 \, S\left[-\frac{1}{2\sinh(\lambda L_0)}\Sigma\,,\Sigma\right] = \frac{1}{2}\left\langle \frac{1+\coth(\lambda L_0)}{2}\Sigma\,,\, Q_B\,\frac{1+\coth(\lambda L_0)}{2}\Sigma\right\rangle \qquad (3.26)$$

$$+\frac{1}{3}\left\langle \frac{1+\coth(\lambda L_0)}{2}\Sigma,\, \frac{1+\coth(\lambda L_0)}{2}\Sigma * \frac{1+\coth(\lambda L_0)}{2}\Sigma\right\rangle$$

$$= -g_o^2\,\widetilde{S}\left[\frac{1+\coth(\lambda L_0)}{2}\,\Sigma\right]\,,$$

after a simple algebra. Notice this action is perfectly well-defined thanks to $\Sigma$ being outside of $\ker L_0$. Again, we observe this is just the action for the Witten's theory after a field redefinition. Not surprisingly, the gauge symmetry also reduces to

$$\Sigma \to \Sigma + Q_B\Lambda + \frac{2}{1+\coth(\lambda L_0)}\left[\frac{1+\coth(\lambda L_0)}{2}\,\Sigma,\, \frac{1+\coth(\lambda L_0)}{2}\,\Lambda\right]\,, \qquad (3.27)$$

and this is equal to the gauge symmetry of the Witten's theory (2.7) after redefining the gauge parameter to $\Lambda \to 2/(1+\coth(\lambda L_0))\Lambda$.

Our analysis above demonstrates that there is no difference between integrating out the string fields $\Psi$ or $\Sigma$ non-perturbatively—both of them leads to Witten's theory. Naively, this is a surprising result. However, since we use (3.18) in either case to integrate the fields out, it is somewhat of a triviality why we obtain the same theory up to a field redefinition. We finally note that it is possible to integrate out the string field $\Psi$ perturbatively plugging (3.14a) to the action (3.1). Instead of repeating the perturbative procedure described in previous subsection, we shall perform this using homotopy transfer in the next section for the sake of brevity.

## 4 Algebraic structure

In this section we describe the underlying $A_\infty$ structure of the action (3.1). We show that the action with the auxiliary field (3.1) endows a differential graded *strictly* associative cyclic algebra, akin to the Witten's theory. We use this underlying algebraic structure and the homological perturbation lemma to complete our discussion on integrating out fields perturbatively. For more details on the homological perturbation theory in the context of SFT, we refer reader to [34, 69–71]. For the discussion on systematically adding stubs to any theory based on an $A_\infty$ algebra see appendix A.

We begin by rewriting the action (3.1) in the following form

$$S[\Phi] = -\text{tr}\left[\frac{1}{2}\langle\Phi, V_1(\Phi)\rangle' + \frac{g_o}{3}\langle\Phi, V_2(\Phi, \Phi)\rangle'\right],\tag{4.1}$$

after scaling $\Psi \to g_o\Psi$ and $\Sigma \to g_o\Sigma$. Here we have defined

$$\Phi \equiv \begin{bmatrix} \Psi \\ \Sigma \end{bmatrix} \in \mathcal{H} \times \mathcal{H} \equiv \mathcal{H}', \quad \langle\Phi_1, \Phi_2\rangle' \equiv \begin{bmatrix} \langle\Psi_1, \Psi_2\rangle & 0 \\ 0 & \left\langle\Sigma_1, \frac{1}{1-e^{-2\lambda L_0}}\Sigma_2\right\rangle \end{bmatrix},\tag{4.2a}$$

$$V_1(\Phi) \equiv \begin{bmatrix} Q_B\Psi \\ Q_B\Sigma \end{bmatrix}, \quad V_2(\Phi_1, \Phi_2) \equiv \begin{bmatrix} e^{-\lambda L_0}\left\{\left(\Sigma_1 - e^{-\lambda L_0}\Psi_1\right) * \left(\Sigma_2 - e^{-\lambda L_0}\Psi_2\right)\right\} \\ -(1-e^{-2\lambda L_0})\left\{\left(\Sigma_1 - e^{-\lambda L_0}\Psi_1\right) * \left(\Sigma_2 - e^{-\lambda L_0}\Psi_2\right)\right\} \end{bmatrix}.\tag{4.2b}$$

We point out the products $V_1, V_2$ are multi-linear maps on $\mathcal{H}'$ and we take the trace over the factors of the doubled space $\mathcal{H}' = \mathcal{H} \times \mathcal{H}$. Again, we take $\Sigma \in (\ker L_0)^c$ implicitly for technical reasons. Our primary claim is that the collection $\mathcal{A}_2 = (\mathcal{H}', V_1, V_2, \text{tr}\langle\cdot, \cdot\rangle')$ forms a differential graded strictly associative cyclic algebra, whose action is given by (4.1). Let us demonstrate this is indeed the case by discussing various aspects of $\mathcal{A}_2$:

- **Grading.** The algebra $\mathcal{A}_2$ is graded by $\mathbb{Z} \times \mathbb{Z}$, combining the ghost number grading of two factors of the Hilbert space $\mathcal{H}'$. The same applies for Grassmann grading and we define

$$(-1)^{|\Phi|} \equiv \begin{bmatrix} (-1)^{|\Psi|} & 0 \\ 0 & (-1)^{|\Sigma|} \end{bmatrix}.\tag{4.3}$$

- **Nilpotency.** This is true given $V_1^2 \propto Q_B^2 = 0$. $V_1$ still increases the ghost numbers by one.

- **Derivation.** This is the property that $V_1$ acts as a derivation on $V_2$. That is for $\Phi_1, \Phi_2 \in \mathcal{H}'$

$$V_1(V_2(\Phi_1, \Phi_2)) = V_2(V_1(\Phi_1), \Phi_2) + V_2((-1)^{|\Phi_1|}\Phi_1, V_1(\Phi_2)).\tag{4.4}$$

Indeed, this trivially follows from the properties of the star product itself and $[Q_B, L_0] = 0$. We point out the binary operation $V_2$, like the star product, does not change the ghost numbers. Notice it was crucial to define (4.3) and place it inside the argument of $V_2$ for this identity.

- **Associativity.** This is the property that, for every $\Phi_1, \Phi_2, \Phi_3 \in \mathcal{H}'$,

$$V_2(V_2(\Phi_1, \Phi_2), \Phi_3) = V_2(\Phi_1, V_2(\Phi_2, \Phi_3)).\tag{4.5}$$

We can demonstrate this is the case by directly evaluating

$$V_2(V_2(\Phi_1, \Phi_2), \Phi_3) = V_2\left(\begin{bmatrix} e^{-\lambda L_0}\left\{\left(\Sigma_1 - e^{-\lambda L_0}\Psi_1\right) * \left(\Sigma_2 - e^{-\lambda L_0}\Psi_2\right)\right\} \\ -(1-e^{-2\lambda L_0})\left\{\left(\Sigma_1 - e^{-\lambda L_0}\Psi_1\right) * \left(\Sigma_2 - e^{-\lambda L_0}\Psi_2\right)\right\} \end{bmatrix}, \Phi_3\right)$$

$$= \begin{bmatrix} -e^{-\lambda L_0}\left\{\left(\Sigma_1 - e^{-\lambda L_0}\Psi_1\right) * \left(\Sigma_2 - e^{-\lambda L_0}\Psi_2\right) * \left(\Sigma_3 - e^{-\lambda L_0}\Psi_3\right)\right\} \\ (1-e^{-2\lambda L_0})\left\{\left(\Sigma_1 - e^{-\lambda L_0}\Psi_1\right) * \left(\Sigma_2 - e^{-\lambda L_0}\Psi_2\right) * \left(\Sigma_3 - e^{-\lambda L_0}\Psi_3\right)\right\} \end{bmatrix}\tag{4.6}$$

$$= V_2\left(\Phi_1, \begin{bmatrix} e^{-\lambda L_0}\left\{\left(\Sigma_2 - e^{-\lambda L_0}\Psi_2\right) * \left(\Sigma_3 - e^{-\lambda L_0}\Psi_3\right)\right\} \\ -(1-e^{-2\lambda L_0})\left\{\left(\Sigma_2 - e^{-\lambda L_0}\Psi_2\right) * \left(\Sigma_3 - e^{-\lambda L_0}\Psi_3\right)\right\} \end{bmatrix}\right) = V_2(\Phi_1, V_2(\Phi_2, \Phi_3)).$$

We remark that the placement of the factors $e^{-\lambda L_0}$ was crucial in order to obtain the second line, after which we used the associativity of the star product.

- Cyclicity. These are the properties

$$\text{tr}\langle\Phi_1, \Phi_2\rangle' = \text{tr}\left[(-1)^{|\Phi_1||\Phi_2|}\langle\Phi_2, \Phi_1\rangle'\right], \tag{4.7a}$$

$$\text{tr}\langle\Phi_1, V_2(\Phi_2, \Phi_3)\rangle' = \text{tr}\left[\langle V_2(\Phi_1, \Phi_2), \Phi_3\rangle'\right], \tag{4.7b}$$

for every $\Phi_1, \Phi_2, \Phi_3 \in \mathcal{H}'$. Again, these follow from the properties of the BPZ product and the fact that $e^{-\lambda L_0}$ is BPZ even (2.6). We remark that the cyclicity here property is slightly different from the usual cyclicity of the Witten's theory given that $(-1)^{|\Phi|}$ is a matrix.

- Non-degeneracy. The bilinear form in (4.2a) is non-degenerate. That is, if (4.2a) vanishes for every $\Phi_1$, then $\Phi_2 = 0$. This follows from the non-degeneracy of the BPZ product and $\ker(1 - e^{-2\lambda L_0})^{-1} = \emptyset$, which in turns follows from the fact that $L_0$ is bounded below for sensible theories. Another way to state this is that the operator $e^{-\lambda L_0}$ is bounded above.

Given such structure, the action (4.1) endows the following gauge symmetry:

$$\Phi \to \Phi + V_1(\Phi) + g_o\left(V_2(\Phi, \Lambda) - V_2(\Lambda, \Phi)\right), \quad \Lambda = \begin{bmatrix} \Lambda_1 \\ \Lambda_2 \end{bmatrix}. \tag{4.8}$$

It is easy to check that this produces (3.2). We remark that generating this gauge transformation from the associative algebra was the motivation behind why we have included the factor $(1 - e^{-2\lambda L_0})^{-1}$ to the inner product and not on the differential $V_1$.

At this point we point out a curious observation that the algebraic structure above only employs the fact that the Grassmann even operator $e^{-\lambda L_0}$ is BPZ even, commutes with $Q_B$, and it is bounded above. So it is possible to replace $e^{-\lambda L_0}$ with any operator $S$ with the same properties to obtain other types of cubic string field theories involving an auxiliary field. This corresponds to adding a generalized version of stub to the cubic vertex, one determined by the operator $S$. We shall elaborate more on this observation in section 5.

## 4.1 Homotopy transfer to the non-polynomial theory

In this subsection we complete our discussion of integrating out $\Sigma$ perturbatively in the sense described in subsection 3.2 and show that the cubic stubbed theory can generate the non-polynomial stubbed theory to all orders. To that end, we use homological perturbation lemma to transfer the algebraic structure of $\mathcal{A}_2$ discussed earlier to the relevant of subspace of $\mathcal{H}'$. We further demonstrate that it is possible to integrate out $\Psi$ perturbatively and $\Sigma$ (or equivalently, $\Psi$) non-perturbatively by transferring this algebraic structure to different subspaces instead.

So, let us begin by defining the symplectic form $\omega : \mathcal{H}' \otimes \mathcal{H}' \to \mathbb{C}$ and the string products

$$\omega(\Phi_1, \Phi_2) \equiv \text{tr}(-1)^{\text{deg}(\Phi_1)}\langle\Phi_1, \Phi_2\rangle', \tag{4.9a}$$

$$m_1(\Phi_1) \equiv V_1(\phi_1), \tag{4.9b}$$

$$m_2(\Phi_1, \Phi_2) \equiv V_2\left((-1)^{\text{deg}(\Phi_1)}\Phi_1, \Phi_2\right). \tag{4.9c}$$

Here deg is the degree for which we have $(-1)^{\text{deg}(\Phi_1)} \equiv (-1)^{|\Phi|+1}$. Unlike $V_1, V_2$, both $m_1, m_2$ are intrinsically degree odd now. We further define the products $m_n = 0$ for $n > 2$.

The advantage of defining $m$-products is that we can now write the relations for the (homotopy) associative algebra in *the tensor coalgebra*

$$T\mathcal{H}' = \bigoplus_{n=0}^{\infty}(\mathcal{H}')^{\otimes n} = \mathbb{C} \oplus \mathcal{H}' \oplus (\mathcal{H}' \otimes \mathcal{H}') \oplus + \cdots, \tag{4.10}$$

compactly in terms of a *coderivation* $\mathbf{m} = \mathbf{m_1} + \boldsymbol{\delta}\mathbf{m}$ that satisfies[6]

$$\mathbf{m}^2 = (\mathbf{m_1} + \boldsymbol{\delta}\mathbf{m})^2 = \mathbf{0} \quad \text{where} \quad \boldsymbol{\delta}\mathbf{m} = \sum_{n=2}^{\infty}\mathbf{m_n} = \mathbf{m_2} + \mathbf{m_3} + \cdots = \mathbf{m_2}, \tag{4.11}$$

where $\mathbf{0}$ is the zero element of $T\mathcal{H}'$ and the individual coderivations $\mathbf{m_n}$ are defined by

$$\mathbf{m_n} = \sum_{i=1}^{\infty}\sum_{j=0}^{i-1}\mathbb{I}^{\otimes j} \otimes m_n \otimes \mathbb{I}^{\otimes(i-j-1)} = m_n + (\mathbb{I} \otimes m_n + m_n \otimes \mathbb{I}) + \cdots. \tag{4.12}$$

Here $\mathbb{I}$ is the identity on $\mathcal{H}'$. We remind that $m_n$ are the multi-linear maps $m_n : (\mathcal{H}')^{\otimes n} \to \mathbb{C}$.

Recall that $\mathcal{H}' = \mathcal{H} \times \mathcal{H}$, and $\Psi$ belongs to the first $\mathcal{H}$ and $\Sigma$ belongs to the second $\mathcal{H}$. So we ought to project the fields onto the first $\mathcal{H}$ factor in order to integrate out $\Sigma$ perturbatively. To that end, define the following projector and the inclusion map on $\mathcal{H}'$[7]

$$p(\Phi) = \begin{bmatrix}\Psi \\ 0\end{bmatrix}, \quad \iota(\Psi) = \begin{bmatrix}\Psi \\ 0\end{bmatrix} \quad \text{for} \quad \Psi \in \mathcal{H} \subset \mathcal{H}', \quad \Phi \in \mathcal{H}' = \mathcal{H} \times \mathcal{H}. \tag{4.13}$$

They clearly satisfy

$$p^2 = p, \quad p\iota = 1, \quad p\,m_1 = m_1 p, \quad \iota\,m_1 = m_1\iota, \tag{4.14}$$

These operators can be promoted to the tensor coalgebra $T\mathcal{H}'$ by

$$\mathbf{p} = \sum_{i=1}^{\infty}p^{\otimes i} = p + p \otimes p + \cdots, \quad \boldsymbol{\iota} = \sum_{i=1}^{\infty}\iota^{\otimes i} = \iota + \iota \otimes \iota + \cdots, \tag{4.15}$$

so that they satisfy

$$\mathbf{p}^2 = \mathbf{p}, \quad \mathbf{p}\boldsymbol{\iota} = \mathbf{1}, \quad \mathbf{p}\mathbf{m_1} = \mathbf{m_1}\mathbf{p}, \quad \boldsymbol{\iota}\mathbf{m_1} = \mathbf{m_1}\boldsymbol{\iota}, \tag{4.16}$$

where $\mathbf{1}$ is the identity of $T\mathcal{H}'$. These maps are *cohomomorphisms* [69, 70].

Now, we need to find a contracting homotopy operator $h$ that leads to the *Hodge-Kodaira decomposition*

$$hm_1 + m_1 h = \iota p - \mathbb{I}. \tag{4.17}$$

in order to form a chain homotopy equivalence and transfer the algebraic structure properly. The choice relevant for our purposes is

$$h(\Phi) = -\begin{bmatrix}0 \\ \Delta\Sigma\end{bmatrix}. \tag{4.18}$$

---

[6]The tensor algebra $T\mathcal{H}'$ is a coalgebra as it is endowed with a comultiplication. A coderivation is a linear map that satisfies co-Leibniz rule. For deeper exposition, definitions, and proofs, see [69, 70].

[7]We always take the inclusion maps to be defined on $\mathcal{H} \subset \mathcal{H}'$, i.e. they are canonical inclusions, and $m_1$ is taken to act as $Q_B$ on this space.

Recall that $\Delta$ is the propagator in the Siegel gauge (2.9). Again, there may be subtleties associated with integrating out $L_0 = 0$ modes and technically we should have been using $\Delta \to \Delta(1 - P_0)$ above, where $P_0$ is a projector onto $\ker L_0$. However, as we shall see, they won't cause any problem at the end so we omit this insertion. Notice the operator $h$ further satisfies the so-called *side conditions*

$$h^2 = 0, \quad h\iota = 0, \quad ph = 0, \tag{4.19}$$

and we can lift them to the tensor coalgebra $T\mathcal{H}'$ by first defining

$$\mathbf{h} = \sum_{i=1}^{\infty} \sum_{j=0}^{i-1} \mathbb{I}^{\otimes j} \otimes h \otimes (\iota p)^{\otimes (i-j-1)} = h + (\mathbb{I} \otimes h + h \otimes \iota p) + \cdots, \tag{4.20}$$

then noticing

$$\mathbf{h}\mathbf{m_1} + \mathbf{m_1}\mathbf{h} = \iota\mathbf{p} - \mathbf{1}, \quad \mathbf{h}^2 = \mathbf{0}, \quad \mathbf{h}\iota = \mathbf{0}, \quad \mathbf{ph} = \mathbf{0}. \tag{4.21}$$

The conditions (4.14) and (4.19) show that we are concerned with a *strong deformation retract*.

Now, by *the homological perturbation lemma*, the subspace $\Psi \in \mathcal{H} \subset \mathcal{H}'$ inherits an $A_\infty$ algebra originating from the associative algebra of $\mathcal{H}'$ after projection by $p$. This algebra is described by the nilpotent perturbed coderivation

$$\mathbf{M} = \mathbf{pm_1}\iota + \mathbf{pm_2} \frac{\mathbf{1}}{\mathbf{1 - hm_2}} \iota \quad \text{with} \quad \mathbf{M}^2 = \mathbf{0}. \tag{4.22}$$

Here the inverse operator is formal and it describes the geometric series on $T\mathcal{H}'$

$$\frac{\mathbf{1}}{\mathbf{1 - hm_2}} = \mathbf{1} + \mathbf{hm_2} + \mathbf{hm_2hm_2} + \cdots. \tag{4.23}$$

By defining the projection operators $\pi_n : T\mathcal{H} \to T\mathcal{H}$ such that $\pi_n T\mathcal{H} = \mathcal{H}^{\otimes n}$ we can read the multi-linear $A_\infty$ maps on $\mathcal{H}$ through

$$M_n = \pi_1 \mathbf{M} \pi_n. \tag{4.24}$$

For example, it is somewhat straight-forward to show

$$M_2 = \pi_1 \mathbf{M} \pi_2 = \pi_1 \mathbf{pm_2}\iota\pi_2 = pm_2(\iota\cdot, \iota\cdot), \tag{4.25a}$$

$$M_3 = \pi_1 \mathbf{M} \pi_3 = \pi_1 \mathbf{pm_2hm_2}\iota\pi_3 = pm_2(\mathbb{I} \otimes h + h \otimes (\iota p))(m_2 \otimes \mathbb{I} + \mathbb{I} \otimes m_2)\iota^{\otimes 3}) \tag{4.25b}$$

$$= pm_2(\mathbb{I} \otimes h + h \otimes (\iota p))(m_2(\iota\cdot, \iota\cdot) \otimes \iota + \iota \otimes m_2(\iota\cdot, \iota\cdot)))$$

$$= pm_2(hm_2(\iota\cdot, \iota\cdot) \otimes \iota + \iota \otimes hm_2(\iota\cdot, \iota\cdot)),$$

using (4.14) and (4.19). These are consistent with what we had earlier in (3.25) upon plugging the expressions for $p$ and $h$. We can repeat the analogous computations to evaluate any $M_n$.

In [72], it has been shown that the string products $M_n$ correspond to the sum of all full binary trees with $n$ leafs for which the inputs correspond to leafs and the output corresponds to the root. Under this dictionary one associates leafs with $\iota$, the internal edges with $h$, vertices with the product $m_2$, and the root with $p$. Upon this identification we see that we have obtained the same products as [37]—the only difference being that where we have placed the operator $1 - e^{-2\lambda L_0}$ in the binary

trees. For us, this was part of the 2-string product $m_2$, hence it was living at the vertices of the trees, while this piece was living at the internal edges in [37]. But given that it appears as an overall multiplier in the 2-product (4.2b) and the form of $h$ in (4.18) we can move this piece up to the tree, i.e. to the internal edge connecting the product; see (4.25b) for an example. Since $h = -\Delta = -b_0/L_0$ for us at the internal edges, we reduce to the situation described in [37] after this rearrangement. This also shows that we didn't have to worry about the subtleties associated with the $L_0 = 0$ modes since $L_0$ in the perturbed products $M_n$ always appear in combination $(1 - e^{-2\lambda L_0})\Delta$, which is regular as $L_0 \to 0$, see (2.17). An alternative argument for the equivalence of $M_n$ with those of [37] completely based on the algebraic structure is given in appendix A.1.

To summarize, we show that it is possible to obtain the non-polynomial stubbed theory from the cubic stubbed theory after transferring homotopy to the relevant subspace of $\mathcal{H}'$. The proof for the cyclicity of $M_n$ under the symplectic form works exactly like in [37]. Moreover, given that we have the space $\mathcal{H}' = \mathcal{H} \times \mathcal{H}$, we can consider transferring homotopy to its different subspaces. For instance, we can project the theory to the second $\mathcal{H}$ factor in $\mathcal{H}'$ using

$$\widehat{p}(\Phi) = \begin{bmatrix} 0 \\ \Sigma \end{bmatrix}, \quad \widehat{\iota}(\Sigma) = \begin{bmatrix} 0 \\ \Sigma \end{bmatrix}, \quad \widehat{h}(\Phi) = -\begin{bmatrix} \Delta(1 - P_0)\Psi \\ 0 \end{bmatrix}, \quad (4.26)$$

and this procedure perturbatively integrates out $\Psi$, leaving $\Sigma$. It is trivial to show that these satisfy the analogous relations to (4.14) and (4.19) and the resulting string products can be described using full binary trees again. We have included the projector $1 - P_0$ to $\widehat{h}$ for a good measure since the zero modes of the field $\Psi$ cannot be integrated out to preserve the cohomology, as emphasized in [69].

Regardless, we find the resulting action to be

$$\widehat{S}[\Sigma] = -\frac{1}{2}\langle\Sigma, \frac{Q_B}{1 - e^{-2\lambda L_0}}\Sigma\rangle - \frac{g_o}{3}\langle\Sigma, \Sigma * \Sigma\rangle + \frac{g_o^2}{2}\langle\Sigma * \Sigma, b_0\frac{e^{-2\lambda L_0}}{L_0}(1 - P_0)(\Sigma * \Sigma)\rangle + \cdots, \quad (4.27)$$

after transferring homotopy. Of course, the zero modes coming from $\Psi$ are also present as we don't project them out and they couple to the field $\Sigma$. However, we can safely lift them out of the spectrum at generic momenta [32, 67, 73, 74].

The action (4.27) is quite intriguing. It is well-known adding stubs to the Witten's theory suppresses contributions from higher $L_0$ modes. On the other hand, the action (4.27) mostly gets contributions from the *higher* $L_0$ modes while suppressing the physics of the lighter $L_0$ modes, thanks to the kinetic term and the projector $P_0$. This formulation of open strings deserves more investigation and one may use it to probe the non-perturbative physics of open strings better in this parametrization. For example, it could be interesting to explore the mean-field approximation in the context of SFT using the action (4.27) since the Hubbard-Stratonovich transformation is the initial step for it.

## 4.2 Homotopy transfer to the Witten's theory

It is also possible to integrate out a certain combination of string fields to recover the Witten's theory as we have already observed in previous section. This translates projecting to a specific subspace of $\mathcal{H}'$ in the language of homological perturbation theory. To keep the discussion general,

let us consider transferring homotopy to the general subspace of $\mathcal{H}'$ defined by the "line"

$$\Sigma = K\Psi, \tag{4.28}$$

where $K$ is some BPZ even operator that commutes with $Q_B$, $[Q_B, K] = 0$. The canonical projection and inclusion to this subspace is given by

$$P(\Phi) = \frac{1}{1 + K^2} \begin{bmatrix} 1 & K \\ K & K^2 \end{bmatrix} \begin{bmatrix} \Psi \\ \Sigma \end{bmatrix}, \quad I(\psi) = \begin{bmatrix} 1 \\ K \end{bmatrix} \psi. \tag{4.29}$$

They satisfy $P^2 = P$, $PI = 1$, $Pm_1 = m_1 P$, $Im_1 = m_1 I$. Furthermore, we have

$$H(\Phi) = -\frac{\widetilde{\Delta}}{1 + K^2} \begin{bmatrix} K^2 & -K \\ -K & 1 \end{bmatrix} \begin{bmatrix} \Psi \\ \Sigma \end{bmatrix} \quad \text{with} \quad Hm_1 + m_1 H = IP - \mathbb{I}. \tag{4.30}$$

It is apparent that the side conditions are satisfied, i.e. $H^2 = 0, HI = 0, PH = 0$. upon assuming the operator $\widetilde{\Delta}$ is Grassmann odd with the properties $\{Q_B, \widetilde{\Delta}\} = 1$ and $[K, \widetilde{\Delta}] = 0$.

Like before, it is possible to use the tensor coalgebra $T\mathcal{H}'$ to transfer the homotopy to the subspace defined by (4.28). The algebraic structure on (4.28) after this projection is described by the following nilpotent coderivation

$$\widetilde{\mathbf{M}} = \mathbf{P}\mathbf{m_1}\mathbf{I} + \mathbf{P}\mathbf{m_2} \frac{1}{1 - \mathbf{H}\mathbf{m_2}} \mathbf{I} \quad \text{with} \quad \widetilde{\mathbf{M}}^2 = \mathbf{0}. \tag{4.31}$$

We denoted the lifts of the operators to the tensor coalgebra $T\mathcal{H}'$ with bold letters as usual.

Now, we investigate different choices for $K$. Trivially, if we take $K = 0$, $\widetilde{\Delta} = \Delta$ or $K = \infty$, $\widetilde{\Delta} = \Delta(1 - P_0)$ we reduce to the subspaces considered in the previous subsection. If we would like to reduce to the Witten's theory, we should take $K = -2\sinh(\lambda L_0)$ and $\widetilde{\Delta} = \Delta$. This is motivated by the relation (3.18).[8] Doing so leads to a simplification in the homological perturbation lemma such that the higher product $\widetilde{M}_n$ for $n > 2$ are absent. This is because

$$K = -2\sinh(\lambda L_0) \quad \Longrightarrow \quad \forall\, \Phi_1, \Phi_2 \in \mathcal{H}' \quad Hm_2(\Phi_1, \Phi_2) = 0 \quad \Longrightarrow \quad \mathbf{H}\mathbf{m_2} = \mathbf{0}. \tag{4.32}$$

in (4.31) effectively. Notice the third line follows because $HI = 0$. As a result, the new products can be obtained from

$$\widetilde{\mathbf{M}} = \mathbf{P}\left(\mathbf{m_1} + \mathbf{m_2}\right)\mathbf{I}. \tag{4.33}$$

and this clearly produces the Witten's theory up to a field redefinition. This argument is manifestly independent of $\Psi$ and $\Sigma$—it was only concerned with the combination (4.28), consistent with our earlier discussion on why integrating $\Psi$ and $\Sigma$ non-perturbatively has lead to the same theory.

## 4.3  A comment on solutions and actions

In this subsection we remark on the solutions of the cubic theory with the auxiliary field (4.1) and how they are mapped to the solutions of the theories resulting after projections. We also remark on the actions of the resulting theories; in particular we show the equality of their on-shell actions.

---

[8]We don't have to include the projector $1 - P_0$ to $H$ as the identity (4.32) holds for any $L_0$.

Unless specified otherwise, the tilde on the quantities refers to *perturbed* operators after projecting to the subspaces of the form (4.28) in this subsection. Our arguments are in the spirit of [69] where similar analysis have performed in the context of the effective actions, so we are going to be brief in our exposition.

We begin our discussion by promoting the equation of motion associated with (4.1) to the tensor coalgebra $T\mathcal{H}'$. It is given using *group-like element* in $T\mathcal{H}'$ by

$$\mathbf{m} \, \frac{1}{1-\Phi} = \mathbf{0} \quad \text{where} \quad \frac{1}{1-\Phi} = \sum_{n=0}^{\infty} \Phi^{\otimes n} = 1 + \Phi + \Phi \otimes \Phi + \cdots. \tag{4.34}$$

Analogously, the equation of motion after transferring homotopy to the theory corresponding to the subspace (4.28) is given by

$$\widetilde{\mathbf{M}} \, \frac{1}{1-\Phi} = \mathbf{0}. \tag{4.35}$$

Now assume $\Phi = \Phi^*$ solves $(4.34)$[9] and consider the string field

$$\widetilde{\Phi^*} = \pi_1 \widetilde{\mathbf{P}} \frac{1}{1-\Phi^*}, \tag{4.36}$$

after projection, where $\widetilde{\mathbf{P}}$ is the perturbed projector

$$\widetilde{\mathbf{P}} = \mathbf{P} \, \frac{1}{1 - \mathbf{m_2}\,\mathbf{H}} \quad \text{where} \quad \widetilde{\mathbf{P}}\,\mathbf{m} = \widetilde{\mathbf{M}}\,\widetilde{\mathbf{P}}. \tag{4.37}$$

We claim $\widetilde{\Phi^*}$ solves the equation of motion (4.35). This can be shown by

$$\widetilde{\mathbf{M}} \, \frac{1}{1-\widetilde{\Phi^*}} = \widetilde{\mathbf{M}} \, \frac{1}{1 - \pi_1\widetilde{\mathbf{P}}\frac{1}{1-\Phi^*}} = \widetilde{\mathbf{M}}\widetilde{\mathbf{P}} \, \frac{1}{1-\Phi^*} = \widetilde{\mathbf{P}}\,\mathbf{m} \, \frac{1}{1-\Phi^*} = 0, \tag{4.38}$$

where we have used the fact $\widetilde{\mathbf{P}}$ is a cohomomorphism on the tensor coalgebra and a subsequent identity for it [69]. Notice the similarity of our arguments to [37]. The only difference here being that we can directly use the perturbed projector $\widetilde{\mathbf{P}}$ in our argument above; as we are working in the context of strong deformation retract and it is a cohomomorphism. We point out that upon choosing $K = -2\sinh(\lambda L_0)$ to project to the Witten's theory, equation (4.36) produces a solution to the Witten's theory. It is an interesting question whether one can find a previously unknown solutions to the Witten's theory and/or cure the problems associated with the singular solutions (such as the so-called identity-based solutions [13, 76]) by solving the cubic stubbed theory first. These points deserve further investigation.

Similarly, we can show any solution to the projected theory $\psi = \psi^*$ can be lifted to the solution to the cubic stubbed theory by

$$\Phi^* = \pi_1 \widetilde{\mathbf{I}} \, \frac{1}{1-\psi^*}, \tag{4.39}$$

using the so-called perturbed inclusion

$$\widetilde{\mathbf{I}} = \frac{1}{1 - \mathbf{H}\,\mathbf{m_2}}\mathbf{I} \quad \text{where} \quad \widetilde{\mathbf{I}}\,\widetilde{\mathbf{M}} = \mathbf{m}\,\widetilde{\mathbf{I}}. \tag{4.40}$$

---

[9]We warn reader that we haven't solved our theory (4.1). Although it is not hard to imagine that it can be solved directly given its cubic nature.

The reasoning is similar to before:

$$\mathbf{m}\,\frac{1}{1-\Phi^*} = \mathbf{m}\,\frac{1}{1-\pi_1\widetilde{\mathbf{I}}\frac{1}{1-\widetilde{\psi^*}}} = \mathbf{m}\,\widetilde{\mathbf{I}}\,\frac{1}{1-\psi^*} = \widetilde{\mathbf{I}}\,\widetilde{\mathbf{M}}\,\frac{1}{1-\psi^*} = 0\,. \qquad (4.41)$$

In particular we can lift solutions to the Witten's theory to the cubic stubbed theory this way. We remark that promoting solutions to $\mathcal{H}'$ then projecting to the subspace we always end up with the same solution, given $\widetilde{\mathbf{P}}\widetilde{\mathbf{I}} = \mathbf{1}$. Achieving the same for the other way is not guaranteed a priori.

Notice we can first promote the solution $\Psi^*$ of the Witten's theory to the cubic stubbed theory, then project it to the solution $\widetilde{\Phi}^*$ of the non-polynomial stubbed theory and subsequently obtain

$$\widetilde{\Phi}^* = \pi_1\widetilde{\mathbf{p}}\,\frac{1}{1-\pi_1\widetilde{\mathbf{I}}\frac{1}{1-\Psi^*}} = \pi_1\widetilde{\mathbf{p}}\,\widetilde{\mathbf{I}}\,\frac{1}{1-\Psi^*} = \pi_1\mathbf{G}\,\frac{1}{1-\Psi^*}\,. \qquad (4.42)$$

Here we take $\widetilde{\mathbf{I}}$ to be the perturbed inclusion of the Witten's theory to cubic stubbed theory (i.e. $K = -2\sinh(\lambda L_0)$ in (4.29)) and $\widetilde{\mathbf{p}}$ is the perturbed projector to the non-polynomial stubbed theory. We have defined $\mathbf{G} \equiv \widetilde{\mathbf{p}}\,\widetilde{\mathbf{I}}$ above. Notice it is a cohomomorphism by being a product of two cohomomorphisms. We comment more on the significance of $\mathbf{G}$ below.

Given that we have two solutions in two theories we can compare their actions and show they are related. In particular, on-shell actions are observable and they are related to the tension of the unstable D-brane, thus they are better be equal. So, let us begin writing the action for the cubic stubbed theory in the tensor coalgebra language as

$$S[\Phi] = \int_0^1 dt\, \omega\left(\pi_1\boldsymbol{\partial_t}\frac{1}{1-\Phi(t)},\pi_1\mathbf{m}\frac{1}{1-\Phi(t)}\right)\,, \qquad (4.43)$$

where $\Phi(t)$ is a smooth deformation from $\Phi(0) = 0$ to $\Phi(1) = \Phi$ and $\boldsymbol{\partial_t}$ is the coderivation associated with the derivative $\partial_t$ in the tensor coalgebra. Similarly, the action for the theory after projection is given by

$$\widetilde{S}[\Phi] = \int_0^1 dt\, \widetilde{\omega}\left(\pi_1\boldsymbol{\partial_t}\frac{1}{1-\Phi(t)},\pi_1\widetilde{\mathbf{M}}\frac{1}{1-\Phi(t)}\right)\,. \qquad (4.44)$$

Here $\widetilde{\omega}$ is defined by the relation

$$\langle\widetilde{\omega}|\,\pi_2 \equiv \langle\omega|\pi_2\mathbf{I} \quad \text{where} \quad \forall\,\Phi_1, \Phi_2 \in \mathcal{H}' \quad \langle\omega|\Phi_1\otimes\Phi_2 \equiv \omega(\Phi_1,\Phi_2)\,, \qquad (4.45)$$

The products associated with $\widetilde{\mathbf{M}}$ are cyclic with respect to the symplectic form $\langle\widetilde{\omega}|$ [69]. That is

$$\langle\widetilde{\omega}|\pi_2\widetilde{\mathbf{M}}\pi_n = 0 \quad \text{for} \quad n \geq 2\,. \qquad (4.46)$$

Notice this provides an alternative argument for the cyclicity of the string products of the non-polynomial theory [37]. Importantly, we point out having altered grading doesn't affect any of the arguments on cyclicity in [69].

We would like to first demonstrate

$$S[\Phi(\psi)] = \widetilde{S}[\psi] \quad \text{where} \quad \frac{1}{1-\Phi(t)} = \widetilde{\mathbf{I}}\,\frac{1}{1-\psi(t)}\,. \qquad (4.47)$$

Here we are not just interested in solutions like in (4.39), but general field configurations, for which we relate their group-like elements as shown. We have already picked a particular smooth interpolation to relate to them. This is always possible, see [69]. Upon inserting the relation into the action (4.43) we see

$$S[\Phi(\psi)] = \int\limits_0^1 dt\, \omega\left(\pi_1\widetilde{\mathbf{I}}\,\partial_t\frac{1}{1-\psi(t)}, \pi_1\widetilde{\mathbf{I}}\widetilde{\mathbf{M}}\frac{1}{1-\psi(t)}\right) = \widetilde{S}[\psi]. \tag{4.48}$$

For the first equality we have used $\widetilde{\mathbf{I}}\partial_t = \partial_t\widetilde{\mathbf{I}}$ and (4.40). In the second equality, we have used the fact $\widetilde{\mathbf{I}}$ is a cyclic cohomomorphism [77, 78]

$$\langle\widetilde{\omega}|\pi_2 = \langle\omega|\pi_2\widetilde{\mathbf{I}}, \tag{4.49}$$

which subsequently implies that it is possible to get rid of $\widetilde{\mathbf{I}}$ and replace the symplectic forms. Similar ideas can be used to establish

$$\widetilde{S}[\widetilde{\Phi}(\Phi)] = S[\Phi] \qquad \text{where} \qquad \frac{1}{1-\widetilde{\Phi}(t)} = \widetilde{\mathbf{P}}\frac{1}{1-\Phi(t)}. \tag{4.50}$$

We note that establishing cyclicity of the cohomomorphism $\widetilde{\mathbf{P}}$ is a delicate matter. It follows from

$$\langle\widetilde{\omega}|\pi_2\widetilde{\mathbf{P}} = \langle\omega|\pi_2\widetilde{\mathbf{I}}\widetilde{\mathbf{P}} = \langle\omega|\pi_2\left[\widetilde{\mathbf{H}}\mathbf{m} + \mathbf{m}\widetilde{\mathbf{H}} + \mathbf{1}\right] = \langle\omega|\pi_2. \tag{4.51}$$

Here the perturbed $\widetilde{\mathbf{H}}$ is given by

$$\widetilde{\mathbf{H}} = \frac{1}{1-\mathbf{H}\,\mathbf{m_2}}\,\mathbf{H} \qquad \text{where} \qquad \widetilde{\mathbf{H}}\mathbf{m_1} + \mathbf{m_1}\widetilde{\mathbf{H}} = \widetilde{\mathbf{I}}\,\widetilde{\mathbf{P}} - \mathbf{1}. \tag{4.52}$$

In (4.51), we used the cyclicity of $\mathbf{m}$ with respect to $\langle\omega|$ to eliminate the term $\langle\omega|\pi_2\mathbf{m}\widetilde{\mathbf{H}}$. The term $\langle\omega|\pi_2\widetilde{\mathbf{H}}\mathbf{m}$ can be shown to vanish too by repeating the argument around the equation (2.109) in [69]. This completes our analysis of the on-shell actions. In passing, we note that the procedure here is not just limited to on-shell actions, but can be repeated for generic observables [69]. Similarly, one can analyze the fate of the gauge transformations after projections/inclusions.

We finally point out that lifting the Witten's theory $\widetilde{S}$ to the cubic stubbed theory first, then projecting the homotopy to the non-polynomial stubbed theory $S'$, we obtain the following relation between their actions

$$S'[\Psi'] = \widetilde{S}[\Psi] \qquad \text{where} \qquad \Psi' = \pi_1\mathbf{G}\frac{1}{1-\Psi}, \tag{4.53}$$

where $\mathbf{G} = \widetilde{\mathbf{p}}\widetilde{\mathbf{I}}$ as before. This is explicitly given by

$$\mathbf{G} = \widetilde{\mathbf{p}}\widetilde{\mathbf{I}} = \mathbf{p}\,\frac{1}{1-\mathbf{m_2}\,\mathbf{h}}\frac{1}{1-\mathbf{H}\,\mathbf{m_2}}\mathbf{I} = \mathbf{p}\,\frac{1}{1-\mathbf{m_2}\,\mathbf{h}}\mathbf{I}, \tag{4.54}$$

where we have used the magic identity (4.32) for the Witten's theory.

The cohomomorphism $\mathbf{G}$ is supposed to be related to the field redefinition defined by the maps $P_n$ given in equation (63) of [37] by the way it is constructed through homological perturbation lemma. It is interesting to read the first non-trivial term of $\mathbf{G}$ by $G_n = \pi_1\mathbf{G}\pi_n$ to compare our results and check they are consistent. Obviously we have $G_1 = 1$ and we find

$$G_2 = pm_2(I\cdot, hI\cdot) + pm_2(hI\cdot, \iota\cdot), \tag{4.55}$$

in the next order. This explicitly evaluates to

$$G_2(\Psi, \Psi) = -2e^{-\lambda L_0} \left( e^{\lambda L_0} \Psi * \frac{b_0}{L_0} \sinh(\lambda L_0) \Psi \right) - 2e^{-\lambda L_0} \left( \frac{b_0}{L_0} \sinh(\lambda L_0) \Psi * e^{-\lambda L_0} \Psi \right) . \quad (4.56)$$

In both cases we obtain $P_n$ given in equation (63) of [37], up to an overall field redefinition of $\Psi$ by $e^{-\lambda L_0}$. This difference arises due to the way the string field is defined in (4.53) relative to the one defined in equation (59) there. The analysis here can be generalized to higher $G_n$. In fact, we can directly show that these two cohomomorphisms are equal to each other after developing some technology. We refer reader to appendix A.1 for the argument applicable to all orders.

# 5    Generalized stubs

In previous section we noted that it was possible to replace the stub operator $e^{-\lambda L_0}$ with a BPZ and Grassmann even operator $S$ satisfying $[Q_B, S] = 0$ whose spectrum is bounded above without changing any of the results. We call such operator *generalized stub* in this section and explore the consequences of this replacement. Given $S$, we can write down a cubic action[10]

$$-g_o^2 S[\Psi, \Sigma] = \frac{1}{2} \langle \Psi, Q_B \Psi \rangle + \frac{1}{2} \left\langle \Sigma , \frac{Q_B}{1 - S^2} \Sigma \right\rangle - \frac{1}{3} \langle \Sigma - S\Psi, (\Sigma - S\Psi) * (\Sigma - S\Psi) \rangle , \quad (5.1)$$

and the associated gauge symmetry would be

$$\Psi \to \Psi + Q_B \Lambda_1 + S [\Sigma - S\Psi , \Lambda_2 - S\Lambda_1] , \quad (5.2a)$$

$$\Sigma \to \Sigma + Q_B \Lambda_2 - (1 - S^2) [\Sigma - S\Psi , \Lambda_2 - S\Lambda_1] . \quad (5.2b)$$

Like in the case of ordinary stubs $S = e^{-\lambda L_0}$, this theory has an underlying cyclic differential graded strictly associative algebra too. Taking $S = 1$, $\Sigma$ lifts up from the spectrum and we reduce to the conventional Witten's theory. In this section we briefly look at the situation when $S \neq 1, e^{-\lambda L_0}$.

One of the simplest way to construct a BPZ even operator $S$ using the universal ingredients of BCFT is by taking it to be a function of $L_0$, i.e. $S = S(L_0)$, that is bounded above. This situation can't be given a clear geometric interpretation like the ordinary stubs in general, except when it is given by a Laplace transform

$$S(L_0) = \int_0^\infty d\lambda \, s(\lambda) e^{-\lambda L_0} , \quad (5.3)$$

which can be interpreted as a superposition of ordinary stubs of various lengths determined by the function $s(\lambda)$. The ordinary stub of length $\lambda_0$ clearly corresponds to $s(\lambda) = \delta(\lambda - \lambda_0)$.

Let us briefly consider an example for $S$ of this sort that may be of physical interest:

$$S_k(L_0) = \frac{1}{1 + e^{k(L_0 - M)}} . \quad (5.4)$$

This is a logistic function centered at a certain mass scale $L_0 = M > 0$, which is $\approx 1$ below $L_0 \lesssim M$, while $\approx 0$ for $L_0 \gtrsim M$. The parameter $k > 0$ determines the sharpness of the transition: the larger

---

[10]We assume $\Sigma$ and $\Lambda_2$ are in the complement of $\ker(1 - S^2)$ for technical reasons like before.

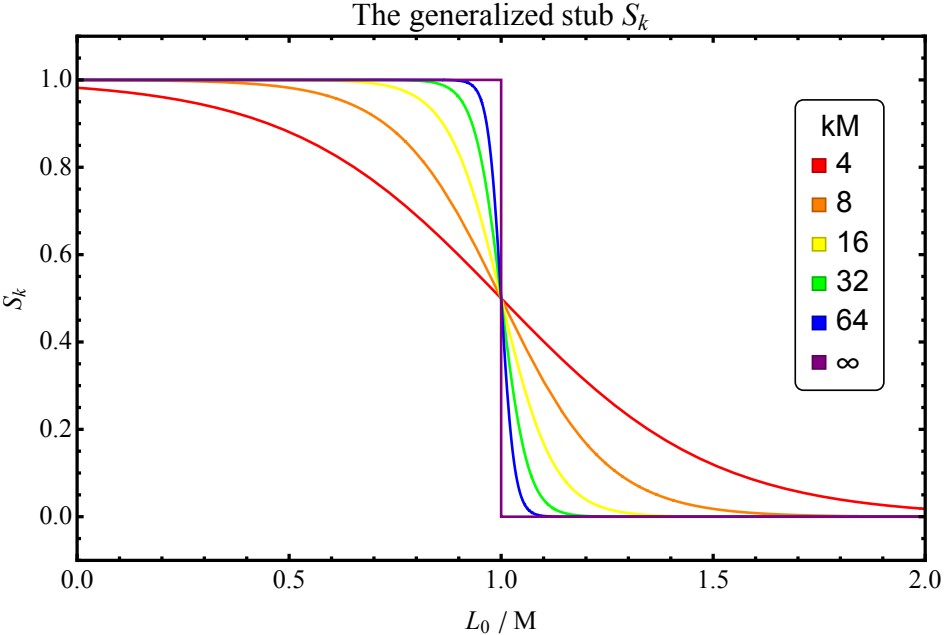

**Figure 3:** The "logistic" stub $S_k(L_0)$ (5.4).

it is the sharper the transition is. Clearly $0 < S_k(L_0) < 1$. Few examples of this function is given in figure 3. Since this function is analytic it is possible to find its inverse Laplace transform and interpret it as a superposition of ordinary stubs, albeit this is complicated and not particularly enlightening from the geometric perspective.

The behavior of $S_k(L_0)$ is similar to the ordinary stub $e^{-kL_0}$ as far as the $L_0 \gg M$ modes are concerned. However, the behavior differs for the $L_0 \lesssim M$ modes. For example, we have argued that the the physics of the higher modes are captured by $\Sigma$ for the ordinary stubs. By using similar logic, we see $\Sigma$ captures the physics of the modes $L_0 \gtrsim M$ in this case, while the rest of the modes is still described by $\Psi$. This suggests integrating out $\Sigma$ perturbatively here is equivalent to integrating out $L_0 \lesssim M$ modes to obtain an effective action for them, i.e. it is possible to incorporate the Wilsonian effective action in our method [79], providing an alternative approach to what has been done in [69].

It is also possible to exclusively keep $L_0 < M$ modes after integrating out $\Sigma$ (i.e. to place a *hard cutoff*) if we consider the $k \to \infty$ limit. This requires considering

$$\lim_{k\to\infty} S_k(L_0) = \Theta(M - L_0) = \frac{1}{2\pi i} \lim_{\epsilon \to 0} \int_{-\infty}^{\infty} d\lambda \, \frac{e^{i(M-L_0)\lambda}}{\lambda - i\epsilon} \,, \tag{5.5}$$

where $\Theta$ is the Heaviside step function. This "step function stub" doesn't have a clear geometric interpretation, given that it is not analytic. However, it may be possible to interpret it as a superposition of "imaginary" stubs via the integral representation in the second line. The extend for which this interpretation is meaningful is unclear however.

Another way to construct the operator $S$ is as follows. Recall that adding stubs to the vertices corresponds to scaling the local coordinates, see (1.1) and figure 1. However, this is not only way to deform the local coordinates. We can also change the original local coordinates $w_i(z)$ to new ones

$w_i'(z)$ by

$$w_i'(z) = f^{-1}(w_i(z)), \quad \text{for} \quad 0 < |w_i'(z)| \le 1, \quad i = 1, \cdots, n. \tag{5.6}$$

Here $f(w')$ is an invertible analytic function on the half-unit disk that satisfies $|f(w')| \le 1$ and maps $w' = 0$ to itself. For example, the relevant function for the ordinary stubs is $f(w) = e^{-\lambda}w$. The function $f(w)$ can be expanded as

$$f(w') = e^{-\lambda}\left[w' + a_2 w'^2 + a_3 w'^3 + \cdots\right]. \tag{5.7}$$

We emphasize that the same function $f$ has to be applied to the local coordinate in order to preserve the cyclic symmetry, hence the covariance, of the theory. Even though this deformation is quite general and applies to all SFTs, we are going to be mostly concerned with the Witten's theory below.

Let us try to understand the operatic meaning of the deformation of the local coordinates by the function $f$ [80–82], which would lead us to the operator $S$. There should be an operator $U_f$ that implements the transformation $f$, which is given by

$$U_f = \exp[v_0 L_0] \exp\left[\sum_{n=1}^{\infty} v_n L_n\right]. \tag{5.8}$$

The negatively-moded Virasoro generators are absent by taking $f(0) = 0$ and we choose to separate the overall scaling. The coefficients $v_n$ can be solved recursively

$$e^{v_0} = f'(0) = e^{-\lambda} \implies v_0 = -\lambda, \tag{5.9a}$$

$$\exp\left[\sum_{n=1}^{\infty} v_n w'^{n+1} \frac{\partial}{\partial w'}\right] w' = \left[w' + a_2 w'^2 + a_3 w'^3 + \cdots\right]. \tag{5.9b}$$

The first few terms are

$$v_1 = a_2, \quad v_2 = -a_2^2 + a_3, \quad v_3 = \frac{3}{2}a_2^3 - \frac{5}{2}a_2 a_3 + a_4. \tag{5.10}$$

It is easy to see the Grassmann even operator $U_f$ commutes with $Q_B$, $[U_f, Q_B] = 0$, and bounded above as a consequence of the constraint $0 \le |f(w')| \le 1$.

We would like to construct a BPZ even operator. By itself, $U_f$ is not BPZ even. A simple way to construct a BPZ even operator is through considering the following combination of $U_f$ with its BPZ conjugate $U_f^{\star}$

$$S = \sqrt{U_f U_f^{\star}}, \quad \text{where} \quad U_f^{\star} = \exp\left[\sum_{n=1}^{\infty}(-1)^n v_n L_{-n}\right]\exp[v_0 L_0], \tag{5.11}$$

via employing the BPZ conjugate of $L_n$, $L_n^{\star} = (-1)^n L_{-n}$ [13]. Again, we can construct even more general BPZ even operators by considering the functions of these $S$'s.

Naively, it may be surprising to have $U_f$ and $U_f^{\star}$ in the stub operator $S$ that is supposed to correspond to the deformations of the local coordinates by $f$. However, this can be heuristically

motivated by reasoning what the stubs in the sliver frame is supposed to look like based on the Schnabl-gauge propagator [8, 83, 84]. Recall this propagator is *formally* given by[11]

$$\mathcal{P} = \frac{\mathcal{B}_0}{\mathcal{L}_0} Q_B \frac{\mathcal{B}_0^{\star}}{\mathcal{L}_0^{\star}}. \tag{5.12}$$

Here $\mathcal{L}_0$ is the zero mode of the stress-energy tensor in the sliver frame

$$\mathcal{L}_0 = L_0 + \sum_{k=1}^{\infty} \frac{2(-1)^{k+1}}{4k^2 - 1} L_{2k} = L_0 + \frac{2}{3} L_2 - \frac{2}{15} L_4 + \cdots, \tag{5.13}$$

and $\mathcal{B}_0$ is the zero mode of the $b$-ghost in the same frame, given in similar way. The inverse of $\mathcal{L}_0$ admits a Schwinger representation

$$\frac{1}{\mathcal{L}_0} = \int_0^{\infty} d\lambda_1 e^{-\lambda_1 \mathcal{L}_0} = \int_0^{\infty} d\lambda_1 \exp\left[-\lambda_1 \left(L_0 + \frac{2}{3} L_2 - \frac{2}{15} L_4 + \cdots\right)\right], \tag{5.14}$$

and we have an analogous expression for $1/\mathcal{L}_0^{\star}$. From these representations and the form of the propagator (5.12), it is a natural expectation that both $U_f$ and $U_f^{\star}$ should be present in the expressions for the (sliver) stubs determined by $f$. Our proposal in (5.11) appears to be consistent with this expectation.

The reasoning above was somewhat heuristic and it is far from complete. We expect such $S$ corresponds adding stubs in other frames up to subtleties that are not apparent in this picture. We plan to investigate the deformations by (5.11) elsewhere. In particular, investigating it for the sliver frame is crucial. It would allow us to lift up the analytic solutions of Witten's theory to the stubbed open SFT and provide an analytic control in a non-polynomial theory as a result. Furthermore, we may obtain an improved understanding of the decomposition of moduli spaces for non-Siegel gauges, such as for the linear $b$-gauges [84], using the stubs of the form (5.11). This may also have important ramifications for closed SFT.

## 6    Conclusion and discussion

In this note we provide a cubic formulation to the stubbed open SFT by introducing an auxiliary string field. We have investigated its gauge symmetries, the equations of motion and the relevant associative algebra and showed that integrating out fields using appropriate combinations of equations of motion recovers the conventional cubic and the non-polynomial stubbed open SFT. We discussed the possible generalization of our construction with the generalized stubs. Our analysis was entirely in the context of strong deformation retract, providing an extension and improvement of the work of Schnabl and Stettinger [37].

Our motivation behind this work was to initiate the study of whether the elementary vertex regions of hyperbolic closed SFT can be decomposed analogously to the non-polynomial stubbed open SFT to achieve a cubic covariant closed SFT [44]. We conclude our paper by listing some similarities and differences between these two cases. Similarities include:

---

[11]We point out it is possible to write the Siegel gauge propagator as $\frac{b_0}{L_0} Q_B \frac{b_0}{L_0}$ and our argument would be consistent with what we have considered throughout the paper.

1. Both theory are equipped with a geometric picture suggesting a decomposition of the Riemann surfaces into cubic ingredients. In the stubbed open SFT this follows from stubs, while in the hyperbolic closed SFT it is due to the pair of pants decomposition [85].

2. In either theory, there are finite-sized geometric objects that is natural to associate with the propagation of auxiliary string fields. In the stubbed open SFT, these were finite-sized strips of length $2\lambda$ as we have argued, while the similar role is expected to be played by the hyperbolic collars around the internal simple closed geodesics in the hyperbolic closed SFT.

We have seen these facts about geometry naturally relate to how the auxiliary string field behaves as far as the stubbed open SFT is concerned. It is likely that the situation may be similar in hyperbolic closed SFT. However, there are few crucial differences that pose immediate puzzles:

1. The moduli for the finite-sized strip are real, while the moduli for the hyperbolic collars are complex due to twists. The physical quantities for the former are analytic functions of the moduli in the vertex region as a result. This is not the case for the latter.

2. The moduli for the hyperbolic collars are constrained by each other in hyperbolic closed SFT. That is, if the width of one hyperbolic collar changes, the widths of the rest of them change as well. This never happens in the stubbed open SFT.

Because of these reasons, it appears to us that the vertex regions resulting from ordinary stubs may be too trivial to capture the true essence of the vertex regions of closed SFT. The extent for which the analytic solutions one has for the stubbed theory (either in the non-polynomial or cubic formulation) are similar to the possible closed SFT solutions is not immediately clear. We leave investigating this problem to future.

# Acknowledgments

We thank Ted Erler, Martin Schnabl, Georg Stettinger, and Barton Zwiebach for their insightful comments on the early draft; Ivo Sachs for suggesting to consider open SFT with stubs; and Tomas Codina, Ted Erler, Olaf Hohm, Manki Kim, Ivo Sachs, Jaroslav Scheinpflug, Martin Schnabl, Ashoke Sen, Georg Stettinger, and Barton Zwiebach for discussions. We are also grateful to the organisers of the Pollica Summer Workshop supported by the Regione Campania, Università degli Studi di Salerno, Università degli Studi di Napoli "Federico II", the Physics Department "Ettore Pancini" and "E.R. Caianiello", and Istituto Nazionale di Fisica Nucleare. AHF further thanks University of Colorado Boulder where the significant portion of this work is completed.

This material is based upon work supported by the U.S. Department of Energy, Office of Science, Office of High Energy Physics of U.S. Department of Energy under grant Contract Number DE-SC0012567. This project has received funding from the European Union's Horizon 2020 research and innovation program under the Marie Sklodowska-Curie grant agreement No 891169.

# A Deformation of a generic $A_\infty$ algebra by stubs

In this appendix we describe the procedure of including (generalized) stubs to a field theory in the framework of homotopy algebras. We do this by considering an arbitrary field theory based on an $A_\infty$ algebra and construct an "enriched" $A_\infty$ algebra with an auxiliary field based on the original one, emphasizing that integrating out this field corresponds including stubs at the end. Since the manipulations are almost equivalent to what has been done for the Witten's theory we keep our discussion brief. The reasoning here can be generalized to $L_\infty$ algebras trivially. Finally, we apply the technology developed here to provide an explicit justification to the method used by Schnabl and Stettinger [37].

So, say we have an arbitrary $A_\infty$ algebra on $\mathcal{H}$ constructed using the multi-linear products $\mathcal{M}_n : \mathcal{H}^{\otimes n} \to \mathcal{H}$. We denote their associated coderivations on the tensor coalgebra $T\mathcal{H}$ given as in (4.12) by $\boldsymbol{\mathcal{M}_n}$. The total coderivation

$$\boldsymbol{\mathcal{M}} = \boldsymbol{\mathcal{M}_1} + \boldsymbol{\delta\mathcal{M}} \qquad \text{where} \qquad \boldsymbol{\delta\mathcal{M}} = \sum_{n=2}^{\infty} \boldsymbol{\mathcal{M}_n} = \boldsymbol{\mathcal{M}_2} + \boldsymbol{\mathcal{M}_3} + \boldsymbol{\mathcal{M}_4} + \cdots, \qquad (A.1)$$

is nilpotent, $\boldsymbol{\mathcal{M}}^2 = \mathbf{0}$. In particular this implies

$$\boldsymbol{\mathcal{M}}^2 = (\boldsymbol{\mathcal{M}_1} + \boldsymbol{\delta\mathcal{M}})^2 = \mathbf{0} \implies \boldsymbol{\mathcal{M}_1}\,\boldsymbol{\delta\mathcal{M}} + \boldsymbol{\delta\mathcal{M}}\,\boldsymbol{\mathcal{M}_1} + \boldsymbol{\delta\mathcal{M}}^2 = \mathbf{0}, \qquad (A.2)$$

given that $\boldsymbol{\mathcal{M}_1}^2 = \mathbf{0}$ by itself.

Our goal is to construct a new $A_\infty$ algebra on the doubled space $\mathcal{H}' = \mathcal{H} \times \mathcal{H}$ upon including an auxiliary field like in (4.2). For convenience, let us define $s : \mathcal{H} \times \mathcal{H} \to \mathcal{H}$ and $t : \mathcal{H} \to \mathcal{H} \times \mathcal{H}$ based on a Grassmann even operator $S$ as follows[12]

$$s = \begin{bmatrix} S & -1 \end{bmatrix}, \qquad t = \begin{bmatrix} S \\ -(1 - S^2) \end{bmatrix}. \qquad (A.3)$$

Notice the property $st = 1$. We demand $S$ satisfies $[\mathcal{M}_1, S] = 0$. If we would like to consider the ordinary stubs in SFT we take $S = e^{-\lambda L_0}$ like before. Using $s$ and $t$, we can construct a cohomomorphism of the form

$$\mathbf{s} = \sum_{i=1}^{\infty} s^{\otimes i} = s + s \otimes s + \cdots, \quad \boldsymbol{t} = \sum_{i=1}^{\infty} t^{\otimes i} = t + t \otimes t + \cdots, \qquad (A.4)$$

acting on the tensor coalgebra $T\mathcal{H}'$. They satisfy $\mathbf{s}\boldsymbol{t} = \mathbf{1}$ as a consequence of $st = 1$.

Now we can construct a coderivation $\boldsymbol{m}$ on $T\mathcal{H}'$ from $\boldsymbol{\mathcal{M}}$ on $T\mathcal{H}$ by defining

$$\boldsymbol{m} \equiv \boldsymbol{\mathcal{M}_1} + \boldsymbol{\Pi}\left[\boldsymbol{t}\,\boldsymbol{\delta\mathcal{M}}\,\mathbf{s}\right]. \qquad (A.5)$$

Here $\boldsymbol{\mathcal{M}_1}$ is assumed to apply both components in $\mathcal{H}'$ as in $\mathcal{H}$ and the linear operator $\boldsymbol{\Pi}$ is a formal object that replaces any combination of $ts$ with the identity $\mathbb{I}$ of $\mathcal{H}'$ in its argument, i.e.

$$\boldsymbol{\Pi}[\cdots \otimes ts \otimes \cdots] = \cdots \otimes \mathbb{I} \otimes \cdots. \qquad (A.6)$$

---

[12]The authors thank Ted Erler for suggesting this trick.

In other words, we pretend the combination $ts$ is $\mathbb{I}$ in the expressions when $\boldsymbol{\Pi}$ is applied. Observe that including $\boldsymbol{\Pi}$ is necessary, otherwise the co-Leibniz rule fails and $\boldsymbol{m}$ can't be a coderivation. This can be easily seen by explicitly writing

$$\boldsymbol{t}\,\boldsymbol{\delta}\boldsymbol{\mathcal{M}}\,\boldsymbol{s} = \sum_{n=2}^{\infty} \boldsymbol{t}\,\boldsymbol{\mathcal{M}_n}\,\boldsymbol{s} = \sum_{n=2}^{\infty}\sum_{i=1}^{\infty}\sum_{j=0}^{i-1}(ts)^{\otimes j} \otimes t\mathcal{M}_n(\underbrace{s\cdot,\cdots,s\cdot}_{n\text{ times}}) \otimes (ts)^{\otimes(i-j-1)}\,, \tag{A.7}$$

and noticing that $\boldsymbol{t}\,\boldsymbol{\delta}\boldsymbol{\mathcal{M}}\,\boldsymbol{s}$ can be made a coderivation only if $ts \to \mathbb{I}$.

The coderivation $\boldsymbol{m}$ is nilpotent as it satisfies

$$\boldsymbol{m}^2 = \boldsymbol{\mathcal{M}_1}^2 + \boldsymbol{\mathcal{M}_1}\,\boldsymbol{\Pi}\,[\boldsymbol{t}\,\boldsymbol{\delta}\boldsymbol{\mathcal{M}}\,\boldsymbol{s}] + \boldsymbol{\Pi}\,[\boldsymbol{t}\,\boldsymbol{\delta}\boldsymbol{\mathcal{M}}\,\boldsymbol{s}]\,\boldsymbol{\mathcal{M}_1} + \boldsymbol{\Pi}\,[\boldsymbol{t}\,\boldsymbol{\delta}\boldsymbol{\mathcal{M}}\,\mathbf{s}]\,\boldsymbol{\Pi}\,[\boldsymbol{t}\,\boldsymbol{\delta}\boldsymbol{\mathcal{M}}\,\mathbf{s}]$$

$$= \boldsymbol{\Pi}\,[\boldsymbol{\mathcal{M}_1}\,\boldsymbol{t}\,\boldsymbol{\delta}\boldsymbol{\mathcal{M}}\,\boldsymbol{s} + \boldsymbol{t}\,\boldsymbol{\delta}\boldsymbol{\mathcal{M}}\,\boldsymbol{s}\,\boldsymbol{\mathcal{M}_1} + \boldsymbol{t}\,\boldsymbol{\delta}\boldsymbol{\mathcal{M}}\,\boldsymbol{s}\,\boldsymbol{t}\,\boldsymbol{\delta}\boldsymbol{\mathcal{M}}\,\mathbf{s}]$$

$$= \boldsymbol{\Pi}\,[\boldsymbol{t}\,(\boldsymbol{\mathcal{M}_1}\,\boldsymbol{\delta}\boldsymbol{\mathcal{M}} + \boldsymbol{\delta}\boldsymbol{\mathcal{M}}\,\boldsymbol{\mathcal{M}_1} + \boldsymbol{\delta}\boldsymbol{\mathcal{M}}^2)\,\boldsymbol{s}] = \boldsymbol{0}\,. \tag{A.8}$$

This manipulation requires some explanation. In the second line we placed $\boldsymbol{\mathcal{M}_1}$ inside $\boldsymbol{\Pi}$ trivially. More importantly, we used

$$\mathbf{s}\,\boldsymbol{\Pi}\,[\boldsymbol{t}\,\boldsymbol{\delta}\boldsymbol{\mathcal{M}}\,\mathbf{s}] = \mathbf{s}\,\boldsymbol{t}\,\boldsymbol{\delta}\boldsymbol{\mathcal{M}}\,\mathbf{s}\,. \tag{A.9}$$

This equality holds true by applying $\mathbf{s}\,\boldsymbol{\Pi}$ and $\mathbf{s}$ to (A.7) and using $st = 1$. Continuing on, in the third line we have commuted $\boldsymbol{t}$ and $\boldsymbol{s}$ with $\boldsymbol{\mathcal{M}_1}$, employed $\mathbf{s}\boldsymbol{t} = \mathbf{1}$ and subsequently used the $A_\infty$ relations of the original string products (A.2).

This reasoning shows that we have defined a new, enriched $A_\infty$ algebra on the doubled space $\mathcal{H}' = \mathcal{H} \times \mathcal{H}$ through (A.5). The individual products can be easily read from (A.7):

$$m_1 = \mathcal{M}_1 \quad \text{and} \quad m_n = t\mathcal{M}_n(\underbrace{s\cdot,\cdots,s\cdot}_{n\text{ times}}) \quad \text{for} \quad n = 2, 3, \cdots\,. \tag{A.10}$$

It is clear that we obtain the products given in (4.2b) when we demand that $\mathcal{M}_2$ is given by the star product of the Witten's theory and $\mathcal{M}_n = 0$ for $n > 2$ after performing a suspension.

Let us discuss the cyclicity properties of the enriched $A_\infty$ algebra now. Suppose the original $A_\infty$ algebra on $\mathcal{H}$ is cyclic. Denote the non-degenerate symplectic form on the $A_\infty$ algebra on $\mathcal{H}$ by $\langle\Omega| : \mathcal{H} \otimes \mathcal{H} \to \mathcal{H}$ and assume this algebra is cyclic under $\langle\Omega|$. That is, for all $n \geq 1$,

$$\langle\Omega|\pi_2\boldsymbol{\mathcal{M}_n} = \langle\Omega|\,(1 \otimes \mathcal{M}_n + \mathcal{M}_n \otimes 1) = 0\,. \tag{A.11}$$

We further demand that $S$ satisfies

$$\langle\Omega|1 \otimes S = \langle\Omega|S \otimes 1\,. \tag{A.12}$$

Clearly, this is true if one takes $S = e^{-\lambda L_0}$ and the symplectic form is based on the BPZ product.

We define the symplectic form $\langle\omega| : \mathcal{H}' \otimes \mathcal{H}' \to \mathbb{C}$ on the enriched algebra to be

$$\langle\omega| \equiv \text{tr}\begin{bmatrix} \langle\Omega| & 0 \\ 0 & \langle\Omega|1 \otimes (1 - S^2)^{-1} \end{bmatrix}\,, \tag{A.13}$$

where we imagine the matrix inside the trace acts on two $\mathcal{H}$ factors of the doubled space $\mathcal{H}' = \mathcal{H} \times \mathcal{H}$. It is easy to show

$$\langle \omega | \Phi_1 \otimes \Phi_2 = -\langle \omega | (-1)^{\deg(\Phi_1)} \Phi_1 \otimes (-1)^{\deg(\Phi_2)} \Phi_2 \,. \tag{A.14}$$

and $\langle \omega |$ is non-degenerate if $S$ is bounded above, after repeating the arguments in section 4.

The multi-linear products $m_n$ (A.5) are cyclic under this symplectic form. For $m_1 = \mathcal{M}_1$ this is obvious. For the rest first observe

$$\langle \omega | \, \mathbb{I} \otimes m_n = \langle \omega | \, \mathbb{I} \otimes t \mathcal{M}_n (\underbrace{s\cdot, \cdots, s\cdot}_{n \text{ times}}) = \langle \Omega | \, s \otimes \mathcal{M}_n (\underbrace{s\cdot, \cdots, s\cdot}_{n \text{ times}}) \,. \tag{A.15}$$

The last equality can be seen by combining the definition of the symplectic form with (A.3) and using (A.12). Similarly we have

$$\langle \omega | \, m_n \otimes \mathbb{I} = \langle \Omega | \, \mathcal{M}_n (\underbrace{s\cdot, \cdots, s\cdot}_{n \text{ times}}) \otimes s \,. \tag{A.16}$$

Adding them up, we see that the multi-linear products $m_n$ are indeed cyclic

$$\langle \omega | \, \pi_2 \, \boldsymbol{m_n} = \langle \omega | \, \pi_2 \, \boldsymbol{\mathcal{M}_n} s = 0 \,, \tag{A.17}$$

and as a result the enriched $A_\infty$ algebra can be made cyclic. Given this construction, it is apparent that one can integrate out the fields in the second factor of $\mathcal{H}' = \mathcal{H} \times \mathcal{H}$ perturbatively to end up with an $A_\infty$ algebra corresponding to the deformation of the theory based on the original cyclic $A_\infty$ on $\mathcal{H}$ by (generalized) stubs. The details are similar to ones given in section 4.

## A.1 Justifying the Schnabl-Stettinger method

As an application of the technology developed above we can show the equivalence of our methods to the those of Schnabl-Stettinger [37] algebraically. This consists of two parts. First, we demonstrate that the products (4.22) are the same as those obtained in equation (45) of [37]. We have already argued for their equivalence in the main text using full binary trees, here we provide an alternative proof based on homotopy algebras. Yet another proof for the equivalence of [37] to the strong deformation retract based on operads can be found in [86]. Second, we show that the cohomomorphism $\boldsymbol{G}$ (4.54) relating the Witten's theory to the non-polynomial stubbed theory is equivalent to the cohomomorphism constructed via equation (62) of [37]. This provides an explicit justification why the Schnabl-Stettinger's method of relating these two theories work, despite not being a strong deformation retract.

So, begin with the strictly associate algebra associated with the Witten's theory on $\mathcal{H}$ and promote it to the tensor coalgebra $T\mathcal{H}$ as in (A.1). We have $\boldsymbol{\delta \mathcal{M}} = \boldsymbol{\mathcal{M}_2}$ for conciseness but this is not strictly necessary for our arguments. We can restate the homotopy transfer formula in (4.22) using the products of the enriched $A_\infty$ algebra $\boldsymbol{m}$ (A.5) as

$$\mathbf{M} = \boldsymbol{\mathcal{M}_1} + \mathbf{p\Pi} \, [t \, \boldsymbol{\delta \mathcal{M}} \, s] \, \frac{\mathbf{1}}{\mathbf{1 - h\Pi} \, [t \, \boldsymbol{\delta \mathcal{M}} \, s]} \, \boldsymbol{\iota} \quad \text{with} \quad \mathbf{M}^2 = \mathbf{0} \,. \tag{A.18}$$

We remind reader $\mathbf{M}$ here is the coderivation associated with the non-polynomial stubbed theory.

Let us make some useful preliminary observations. First, we see

$$\boldsymbol{s\iota} = \boldsymbol{pt} = \boldsymbol{S} \equiv \sum_{i=1}^{\infty} S^{\otimes i} = S + S \otimes S + \cdots , \tag{A.19}$$

which simply follows from the definitions (A.3), (A.4), (4.13), and (4.15). Notice this cohomomorphism acts on $T\mathcal{H}$. Using them we can further demonstrate

$$\boldsymbol{\mathcal{H}} \equiv \boldsymbol{s}\,\boldsymbol{h}\,\boldsymbol{t} = \sum_{i=1}^{\infty} \sum_{j=0}^{i-1} 1^{\otimes j} \otimes (-(1 - S^2)\Delta) \otimes (S^2)^{\otimes(i-j-1)} , \tag{A.20}$$

after using $st = 1$ and the definition (4.20). This also acts on $T\mathcal{H}$.

Finally, it is possible to eliminate $\boldsymbol{\Pi}$ from the coderivation $\mathbf{M}$ and write

$$\mathbf{M} = \boldsymbol{\mathcal{M}_1} + \boldsymbol{p}\,\boldsymbol{t}\,\boldsymbol{\delta\mathcal{M}}\,\boldsymbol{s}\,\frac{1}{1 - \boldsymbol{h}\,\boldsymbol{t}\,\boldsymbol{\delta\mathcal{M}}\,\boldsymbol{s}}\,\boldsymbol{\iota} = \boldsymbol{\mathcal{M}_1} + \boldsymbol{S_p}\,\boldsymbol{\delta\mathcal{M}}\frac{1}{1 - \boldsymbol{\mathcal{H}}\boldsymbol{\delta\mathcal{M}}}\boldsymbol{S_\iota} , \tag{A.21}$$

given that the combination $ts$ can be taken to never appear in the expressions after imposing certain conditions we discuss below. We have used $\boldsymbol{S_p} = \boldsymbol{S_\iota} = \boldsymbol{S}$ above for reasons that are going to be apparent soon. Now we see (A.21) is precisely equation (45) of [37] upon replacing

$$\boldsymbol{D_W} \to \mathbf{M}, \quad \boldsymbol{d_W} \to \boldsymbol{\mathcal{M}_1}, \quad \boldsymbol{p} \to \boldsymbol{S_p}, \quad \boldsymbol{i} \to \boldsymbol{S_\iota}, \quad \boldsymbol{h} \to \boldsymbol{\mathcal{H}}, \quad \boldsymbol{m_2} \to \boldsymbol{\delta\mathcal{M}}, \tag{A.22}$$

from their expressions to ours and taking $S = e^{-\lambda L_0}$.

However this is not sufficient to establish the equivalence: we have to explain the origin of the factor $P_{SDR}$ that imposes the strong deformation retract relations in [37]. This factor originates from eliminating $\boldsymbol{\Pi}$ in (A.21). For example, we have the following term in the expansion of the geometric series in (A.18) at the leading order for which we are supposed to have the equality

$$\boldsymbol{p}\,\boldsymbol{\Pi}\,[\boldsymbol{t}\,\boldsymbol{\delta\mathcal{M}}\,\boldsymbol{s}]\,\boldsymbol{\iota} = \boldsymbol{S_p}\,\boldsymbol{\delta\mathcal{M}}\,\boldsymbol{S_\iota} , \tag{A.23}$$

from (A.21). A quick inspection shows this equality holds only if we pretend $S_p S_\iota = 1$, despite they give $S^2$. We assume the objects with un-bold letters are the multi-linear products on $\mathcal{H}$ that are associated with their bold counterparts on $T\mathcal{H}$. A similar reasoning for the next two terms in the geometric series expansion of (A.18) shows that demanding identities

$$\boldsymbol{p}\,\boldsymbol{\Pi}\,[\boldsymbol{t}\,\boldsymbol{\delta\mathcal{M}}\,\boldsymbol{s}]\,\mathbf{h}\boldsymbol{\Pi}\,[\boldsymbol{t}\,\boldsymbol{\delta\mathcal{M}}\,\boldsymbol{s}]\boldsymbol{\iota} = \boldsymbol{S_p}\,\boldsymbol{\delta\mathcal{M}}\boldsymbol{\mathcal{H}}\boldsymbol{\delta\mathcal{M}}\boldsymbol{S_\iota} , \tag{A.24a}$$

$$\boldsymbol{p}\,\boldsymbol{\Pi}\,[\boldsymbol{t}\,\boldsymbol{\delta\mathcal{M}}\,\boldsymbol{s}]\,\mathbf{h}\boldsymbol{\Pi}\,[\boldsymbol{t}\,\boldsymbol{\delta\mathcal{M}}\,\boldsymbol{s}]\mathbf{h}\boldsymbol{\Pi}\,[\boldsymbol{t}\,\boldsymbol{\delta\mathcal{M}}\,\boldsymbol{s}]\boldsymbol{\iota} = \boldsymbol{S_p}\,\boldsymbol{\delta\mathcal{M}}\boldsymbol{\mathcal{H}}\boldsymbol{\delta\mathcal{M}}\boldsymbol{\mathcal{H}}\boldsymbol{\delta\mathcal{M}}\boldsymbol{S_\iota} , \tag{A.24b}$$

as in (A.21) forces us to pretend the side conditions $\mathcal{H}S_\iota = S_p\mathcal{H} = 0$ and $\mathcal{H}^2 = 0$ (which is already the case here) are satisfied respectively.[13] There aren't any more conditions coming from the higher order terms in the expansion of the geometric series (A.18). So we conclude that it was consistent to take the combination $ts$ to never appear in (A.21) and have the equality there upon demanding these conditions. This completes the justification of why transferring homotopy without having a strong deformation retract (while pretending otherwise) to include stubs wasn't problematic in [37].

---

[13]We also have to replace the sole $S^2$ factors in the summand of (A.20) with $S_\iota S_p$ to generate the definition $\boldsymbol{\mathcal{H}}$ consistent with the replacement rule (A.22), see (4.20).

Next, we justify the validity of using the equation (62) in [37] by showing that it is equivalent to $\boldsymbol{G}$ (4.54). The reasoning is similar to above. Begin by writing $\boldsymbol{G}$ as

$$\mathbf{G} = \widetilde{\mathbf{p}}\,\widetilde{\mathbf{I}} = \mathbf{p}\,\frac{1}{1 - \mathbf{m_2}\,\mathbf{h}}\mathbf{I} = \mathbf{p}\,\frac{1}{1 - \boldsymbol{\Pi}\,[t\,\delta\boldsymbol{\mathcal{M}}\,s]\,\mathbf{h}}\mathbf{I}\,. \tag{A.25}$$

On top of the observations we have done earlier, we notice that

$$\boldsymbol{s}\,\mathbf{h}\,\mathbf{I} = \sum_{i=1}^{\infty}\sum_{j=0}^{i-1}(S^{-1})^{\otimes j}\otimes(S^{-1}-S)\Delta\otimes S^{\otimes(i-j-1)} = \boldsymbol{\mathcal{H}}\boldsymbol{S}^{-1}\,, \tag{A.26}$$

using (4.13), (4.18) and (4.29). This shows

$$\mathbf{G} = S_p\,\frac{1}{1 - \boldsymbol{\delta\mathcal{M}\mathcal{H}}}\,S_\iota{}^{-1}\,, \tag{A.27}$$

upon getting rid of $\boldsymbol{\Pi}$ factors. Again, we have to pretend the side conditions are satisfied here for the consistency of this procedure.

As we mentioned in the paragraph below (4.56), there is a difference in the definition of the string field relative to [37]. This is reflected by the extra $\boldsymbol{S_\iota}^{-1}$ above. Taking it into account, we see $\mathbf{G}$ is same as the cohomomorphism defined by equation (62) of [37] upon replacing

$$\boldsymbol{P}\to\boldsymbol{G}\,,\quad \boldsymbol{p}\to\boldsymbol{S_p}\,,\quad \boldsymbol{h}\to\boldsymbol{\mathcal{H}}\,,\quad \boldsymbol{m_2}\to\boldsymbol{\delta\mathcal{M}}\,, \tag{A.28}$$

from their expressions to ours and setting $S = e^{-\lambda L_0}$. This was precisely what we wanted to show.

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
