# Peer review of "Open string stub as an auxiliary string field"

_SciPost Physics_

## Round 1 · Referee Report · Anonymous (Referee 1) · 2023-12-16

Report

This impressive paper aims to discuss a method of making a non-polynomial string field theory cubic by means of "integrating in" a set of auxiliary fields. Having such a procedure at our disposal would be highly desirable, especially in the case of closed string field theory whose interactions famously do not allow for a cubic formulation in terms of a single dynamical closed string field.

The authors decide to approach this ambitious goal by considering a simpler scenario within the context of open SFT: they start with a non-polynomial theory which arises from the Witten cubic OSFT by endowing the fundamental cubic vertex with stubs. This is a worthwhile endeavor not only for the sake of providing a toy example before attacking the more challenging case of closed SFT: as the authors point out, understanding the structure of the stubbed theory may shed some light on the properties of its classical solutions and how they may regularize some problematic solutions of Witten OSFT.

In particular, by identifying the surface state associated with open string propagation in the stub region, the authors are able to write down a cubic action $S[\Psi,\Sigma]$ for the dynamical string field $\Psi$ and single auxiliary string field $\Sigma$ such that integrating $\Sigma$ out perturbatively in $g_\mathrm{s}$ gives the non-polynomial stubbed theory, thus strictifying its $A_\infty$ structure. Integrating out $\Sigma$ non-perturbatively while fixing a different partial gauge, the authors also demonstrate equivalence of $S[\Psi,\Sigma]$ with the Witten cubic theory. This enables them to construct the field redefinition relating the Witten theory and the non-polynomial stubbed theory, obtaining results equivalent to those of the recent work of Schnabl and Stettinger. Since the authors develop powerful technology to account for the diagrammatic expansion of the interactions using the mathematical language of tensor coalgebras and homological perturbation theory, they are able to demonstrate validity of their claims to all orders. The authors further explore integrating out the dynamical string field $\Psi$ from $S[\Psi,\Sigma]$ (up to cohomology), obtaining an action with interesting properties with regard to non-perturbative physics. By considering more general stub profiles, they also nicely manage to incorporate the Wilsonian effective action story into their paradigm and provide some initial comments on adding stubs in the sliver frame with the hope of achieving analytic control over the classical solutions of the non-polynomial stubbed theory in the future.

Presentation-wise, it is highly commendable that the authors manage to give a very clear and structured exposition of such a complicated and technical matter, maintaining a very good balance between detailed calculations and explanations.

The referee has the following comments, suggestions and questions:

  1. The referee is concerned with the details of the procedure of non-perturbatively integrating out the dynamical string field $\Psi$, as outlined by the authors in Section 3.3. Based on the l.h.s. of (3.26), it seems to be suggested that this is achieved by substituting

    $$ \Psi(\Sigma) = -\frac{1}{2\sinh(\lambda L_0)}\Sigma $$
    into $S[\Psi,\Sigma]$. However, going back to (3.18), this $\Psi(\Sigma)$ appears to represent a solution to the equations of motion $E_\Psi+e^{-\lambda L_0}E_\Sigma$ only up to a possible term belonging to $\mathrm{ker}\,L_0$. That is, the referee believes that the action (3.26) could generally receive additional contributions from the zero-momentum sector. This would appear to break the observed symmetry between integrating the $\Sigma$ and $\Psi$.

  2. This contribution of the zero-modes to the action after integrating out the dynamical string field $\Psi$ is, in fact , mentioned by the authors for the perturbative case (4.27), where they argue that these additional terms may be neglected at generic momentum. The referee thinks that this may turn out to be a dangerous approximation in a number of applications, in particular for the discussion of classical solutions which only excite states from the zero-momentum sector (such as the tachyon vacuum and marginal deformations) from this new perspective.

  3. At the very end of Section 4.2, the authors reiterate their remark on the symmetry between integrating out $\Sigma$ and $\Psi$ from the action $S[\Psi, \Sigma]$, this time using the tensor coalgebra machinery. However, while it seems to the referee that the contracting homotopy operator $H$ introduced in (4.30) indeed has a well-defined action on $\Phi$ (because $\tilde{\Delta}$ is always protected by $K$ when acting on $\Psi$, which might contain $\mathrm{ker}\,L_0$), the same might not be true for an analogous contracting homotopy which one would use to integrate out $\Psi$ instead of $\Sigma$ (essentially exchanging the roles $\Psi \leftrightarrow \Sigma$ and $K \leftrightarrow 1/K$). As a result, one would probably not end up with a cubic action containing purely $\Sigma$, but instead with some additional $\mathrm{ker}\, L_0$ contamination.

  4. Typos:

  5. page 8: "%68" -> "68 %"

Overall, this paper addresses topics which are of high interest and importance to the recent efforts in the community. The referee believes that, subject to the author's addressing the above comments, the paper will make an excellent contribution to the journal.

  • validity: -
  • significance: -
  • originality: -
  • clarity: -
  • formatting: -
  • grammar: -

Author:  Atakan Hilmi Firat  on 2023-12-19  [id 4200]

(in reply to Report 1 on 2023-12-16)

We thank referee for their time and valuable comments. Here our response to their concerns:

  1. As referee pointed out $\Psi (\Sigma) = - (1/\sinh \lambda L_0) \Sigma$ is indeed the solution to the equation of motion $E_\Psi + e^{-\lambda L_0} E_\Sigma$. It is important to notice that, however, this is only true up to possible $Q_B$-exact terms in general. In the arguments sketched between equations (3.15) and (3.18), we eliminated this concern by fixing our gauge partially before integrating out and argued how the residual terms due to $\ker L_0$ won't contribute in this gauge choice . So we think there won't be additional contributions to the new theory due to $\ker L_0$, just as in when we integrate our $\Sigma$ non-perturbatively, and the symmetry between these procedures are still the case. In fact, integrating out $\Psi$ non-perturbatively does not integrate out cohomology and all $\ker L_0$ states remained in the theory, see our third comment below. Finally, it is important to notice the zero momentum states are not relevant, as they don't belong to $\ker L_0$ in general, which is the primary subtlety here.

  2. The authors agree with the assessment that behavior of the modes belonging to $\ker L_0$ from the perspective of the theory described in (4.27) is indeed not entirely clear; for the classical solutions in particular. The resolution most likely requires understanding how solutions work in non-polynomial theories first and how $\ker L_0$ gets transferred to these redefined solutions. We avoid making any claims about this issue at the moment and hope to make further progress in future. We mostly approached our construction from the perspective of ordinary Hubbard-Stratonovich transformation for $\phi^4$ theory; for which when one integrates out $\phi$ after including auxiliary field, one ignores issues associated with on-shell modes. Nonetheless, from the perspective of perturbative amplitudes of the redefined theory, we think that the subtleties associated with $\ker L_0$ modes (which are on-shell modes) may still be dealt with the standard arguments of lifting them at generic momenta. This is what we wanted to mean in our draft and we are going to clarify this point in the next version. Also, we don't think there would be any issues with zero momentum states of this sort in general as they don't belong to $\ker L_0$ as we point out above. It is worth mentioning that $\ker L_0$ modes should be always present in the theory---as integrating them amounts to integrating out the cohomology, which is forbidden physically and mathematically. For example, in the case of transferring homotopy to obtain effective field theory this is made sure to be case by placing momentum cutoffs. In our case for perturbatively integrating $\Psi$ out this can be understood to be the case by the explicit appearance of the projector $P_0$ in the homotopy contracting operator $\hat{h}$ in equation (4.26), where $P_0$ is the projection onto $\ker L_0$.

  3. An important point to notice here is that when we write the relation $\Psi (\Sigma) = - (1/\sinh \lambda L_0) \Sigma$ (which is what the homotopy transfer procedure described in section (4.2) boils down to eventually), it was crucial that $\Sigma$ doesn't contain any modes in $\ker L_0$ in the cubic stubbed theory as that would make this expression ill-defined, please see our comments below equations (3.2) and (3.13). So the action of the homotopy contracting operator is still well-defined when we try to integrate out $\Psi$ non-perturbatively. Moreover, this procedure doesn't integrate our $\ker L_0$ modes from the theory. The easiest way to see this is there are no $\ker L_0$ states in $\Psi(\Sigma)$ due to $\Sigma$ having none initially. So when it is plugged, every state in $\ker L_0$ remains in the theory. Only place they can reside in $\Sigma$ now and the theory is still cubic like the Witten's theory. It appears we implicitly extend what modes $\Sigma$ contain after integrating out in our procedure. We hope to clarify this implicit procedure in the next version of the draft.

  4. Thanks for noticing the typo.

I hope our comments were satisfactory. In the next version of the draft we hope to address concerns of the referee on $\ker L_0$ modes in an explicit manner. Thank you.

---

## Round 1 · Referee Report · Anonymous (Referee 2) · 2024-4-7

Report

This interesting paper describes how a non-cubic structure induced by a non-trivial stub can be made equivalent to a cubic structure with an auxiliary field in the context of bosonic open string field theory.

While the suggestion that introducing auxiliary fields can simplify the structure of string field theory is very intriguing, there are many questions that remain.

Here are a few comments and questions.

  1. Introduction of and manipulations of the auxiliary field that are presented in the paper remain purely classical. It is not yet clear, if introducing an auxiliary field will not render quantum effects ambiguous for example. This is also in relation to the second comment given by the other referee. It will be interesting to investigate quantum effects in the presence of the auxiliary field.

  2. In the context of topological string theories, action of string field theory is known to be cubic. Although, this will not clarify issues concerning the non-polynomial structure of physical closed string field theory, I wonder if authors have any thoughts on whether the same techniques investigated in the paper can be applied to simple closed string field theory as the first step towards generalizing the method studied in the paper.

Overall, this paper made interesting progress that can be potentially very important. I would recommend the publication of this manuscript.

  • validity: top
  • significance: high
  • originality: high
  • clarity: top
  • formatting: -
  • grammar: -

Author:  Atakan Hilmi Firat  on 2024-04-08  [id 4396]

(in reply to Report 2 on 2024-04-07)
Category:
answer to question
pointer to related literature

The authors thank the referee for the comment. Here our responses:

1 - We point out a recent interesting work by Maccaferri et. al. (2403.10471). Our colleagues have described the procedure of deforming quantum bosonic string field theory theories (both closed and open-closed) with stubs and outlined how the auxiliary string field is introduced for these cases. The concerns of the referee are mostly addressed there and they may want to refer there for more details.

2- As far as the knowledge of the authors goes, introducing stubs, or any deformation for that matter, will be trivial for topological string field theories: such as Chern-Simons theory for open strings or Kodaira-Spencer gravity for closed strings. Our expectation stems from the fact that the amplitudes of these theories get contributions entirely from the boundaries of moduli spaces, which makes only the cubic vertex non-trivial in the string field theory action. Furthermore, the string field theory insertions to this cubic vertex don't see local coordinates by simply being zero modes of the strings. So deforming the cubic vertex shouldn't have any effect on them perturbatively in principle. It is a distinct concern whether the similar reasoning applies non-perturbatively, which may be interesting to study.

I hope our comments were satisfactory.

---

## Editorial Decision

resubmitted